



# Implementation of Primary and Secondary Ice Production in EC-Earth3-AerChem: Global Impacts and Insights

Montserrat Costa-Surós[1], María Gonçalves Ageitos[1,2], Marios Chatziparaschos[1,†,‡],
Paraskevi Georgakaki[3,§], Manu Anna Thomas[4], Gilbert Montané Pinto[1], Stelios Myriokefalitakis[5],
Twan van Noije[6], Philippe Le Sager[6], Maria Kanakidou[7,8,9], Athanasios Nenes[8,10], and Carlos Pérez
García-Pando[1,11]

[1]Barcelona Supercomputing Center (BSC), Barcelona, Spain
[2]Department of Project and Construction Engineering, Universitat Politècnica de Catalunya (UPC), Barcelona, Spain
[3]Leipzig Institute for Meteorology, Leipzig University, Leipzig, Germany
[4]Swedish Meteorological and Hydrological Institute (SMHI), Norrköping, Sweden
[5]Institute for Environmental Research and Sustainable Development, National Observatory of Athens (NOA), Athens, Greece
[6]Royal Netherlands Meteorological Institute (KNMI), De Bilt, Netherlands
[7]Environmental Chemical Processes Laboratory (EPCL), University of Crete, Department of Chemistry, Heraklion, Greece
[8]Foundation for Research and Technology, Center for the Study of Air Quality and Climate Change (C-STACC), Institute of
Chemical Engineering Sciences (ICE-HT), Patras, Greece
[9]Institute of Environmental Physics (IUP), University of Bremen, Bremen, Germany
[10]School of Architecture, Civil and Environmental Engineering (ENAC), Laboratory of Atmospheric Processes and their
Impacts (LAPI), Ecole Polytechnique Federale de Lausanne (EPFL), Lausanne, Switzerland
[11]Catalan Institution for Research and Advanced Studies (ICREA), Barcelona, Spain
[†]formerly at: Environmental Chemical Processes Laboratory (EPCL), University of Crete, Department of Chemistry,
Heraklion, Greece
[‡]formerly at: Foundation for Research and Technology, Center for the Study of Air Quality and Climate Change (C-STACC),
Institute of Chemical Engineering Sciences (ICE-HT), Patras, Greece
[§]formerly at: School of Architecture, Civil and Environmental Engineering (ENAC), Laboratory of Atmospheric Processes
and their Impacts (LAPI), Ecole Polytechnique Federale de Lausanne (EPFL), Lausanne, Switzerland

**Correspondence:** Montserrat Costa-Surós (montserrat.costa@bsc.es)

**Abstract.**

Clouds and aerosol–cloud interactions remain major sources of uncertainty in climate projections. Here, we improve the representation of mixed-phase clouds (MPCs) in the EC-Earth3-AerChem Earth System Model by replacing the default temperature-dependent nucleation scheme with a physically based aerosol-sensitive heterogeneous ice nucleation parameterization. This scheme accounts for immersion freezing by K-feldspar, quartz, and marine organic aerosols, and is combined with

a machine-learning-based parameterization of secondary ice production (SIP) to represent ice crystal multiplication processes.

The new configuration improves agreement with global in situ ice nucleating particle (INP) observations and reveals realistic spatial patterns of ice crystal number concentrations (ICNC) across diverse environments. While biases in liquid water path persist, with overestimations in the tropics and underestimations at high latitudes, the aerosol-sensitive primary ice production

scheme increases supercooled liquid water and cloud cover, particularly in the extratropics. Critically, the addition of SIP rebalances the cloud phase by enhancing ICNC in regions with low primary ice formation.





Compared to the default scheme, the aerosol-sensitive primary ice production configuration with SIP reduces cloud radiative effect biases at mid- and high latitudes, while increasing them in the lower latitudes, leading to comparable global biases across configurations. Our results highlight the importance of explicitly representing both aerosol-sensitive nucleation and SIP
for realistic simulations of MPCs and their radiative impacts. Unlike previous schemes, in which ice concentrations depend directly on INPs, the presence of effective SIP enhances ice formation in all MPCs and reduces the sensitivity of ICNC to aerosols, especially at low INP levels.

## 1 Introduction

Clouds play a central role in regulating the Earth's energy budget, acting as key modulators of both incoming solar and outgoing
terrestrial radiation. Changes in cloud properties, whether driven by aerosol-cloud interactions (ACI) or cloud feedbacks, can strongly influence the energy distribution within the climate system (Forster et al., 2021). According to the Fifth Assessment Report of the Intergovernmental Panel on Climate Change (IPCC), clouds and aerosols remain among the largest contributors to uncertainty in estimates of the Earth's radiative balance and future climate change (Boucher et al., 2013). Despite ongoing progress, accurately representing ACI in Earth System Models (ESMs) remains challenging due to complex, sub-grid-scale
nature of key processes (Boucher et al., 2013; Forster et al., 2021). The IPCC's Sixth Assessment Report (Forster et al., 2021) highlighted that substantial advances in cloud process understanding have reduced the uncertainty in cloud feedbacks by about 50%. However, further improvements in ESMs are needed. For instance, EC-Earth3, as other coupled climate models (Hyder et al., 2018), exhibits a cold bias over extensive parts of the Northern Hemisphere (NH) land regions and the Arctic as well as a warm bias in the high-latitudes of the Southern Hemisphere (SH), including Antarctica. These discrepancies are closely linked
to biases in the shortwave (SW) component of the cloud radiative effect (CRE) (Thomas et al., 2019; Döscher et al., 2022).

In this context, mixed-phase clouds (MPCs) play a pivotal role, as their radiative impact is highly sensitive to the partitioning between liquid and ice phases. Accurately capturing this partitioning is essential for understanding phenomena such as Arctic amplification (Tan and Storelvmo, 2019). However, MPCs remain particularly challenging to represent in climate models owing to their complex microphysics, limiting predictability and contributing to uncertainties in the radiative balance (Korolev
and Field, 2008). The supercooled liquid water fraction in MPCs, responsible for much of the cloud reflectivity, is strongly influenced by the formation, growth and precipitation of ice crystals in MPCs. In a warming climate, the transition from ice to liquid in these clouds is expected to increase their albedo, potentially enhancing the negative cloud-phase feedback (Rosenfeld et al., 2014; Vergara-Temprado et al., 2017; Murray et al., 2021). Hofer et al. (2023) even suggests that improving the realism of MPCs representation in models may amplify future climate warming because more accurate simulations of MPCs increase
the present-day shortwave cooling by supercooled liquid water. Under warming, the subsequent reduction of supercooled liquid decreases cloud reflectivity, so the net warming may be larger than in models with less realistic MPCs. Matus and L'Ecuyer (2017) place the net CRE of MPCs at approximately –3.4 W/m$^2$, dominated by a strong SW cooling effect (–8.1 W/m$^2$) partially offset by longwave (LW) warming (+4.7 W/m$^2$). These figures highlight the importance of accurately simulating MPCs when quantifying cloud feedbacks under future climate scenarios.



In the temperature range between 0 and -38 °C, liquid and ice cloud particles can coexist within clouds. However, the formation of ice requires the presence of ice nucleating particles (INPs), which serve as seeds for droplet freezing (DeMott et al., 2003; Murray et al., 2012; Hofer et al., 2023; Chatziparaschos et al., 2023) and play a critical role in MPC development and evolution. Despite their importance, observations have so far been unable to provide clear evidence for the significance of the effective radiative forcing associated with aerosol impacts on MPCs or ice clouds (Forster et al., 2021). While satellite-based

studies tend to support modelling results qualitatively, they have yet to provide robust quantitative constraints for ice clouds (Forster et al., 2021). Moreover, modelling studies show large discrepancies in both the magnitude and sign of aerosol-induced changes in ice and MPCs, underscoring a limited understanding of these processes (Forster et al., 2021; Murray et al., 2021). Quaas et al. (2024) recognizes that changes in mixed-phase and ice clouds are less well understood than those in liquid-phase clouds, leading to low confidence in their representation and feedback strength (Bellouin et al., 2019).

ACI in MPCs are especially complex due to uncertainties surrounding INP sources, concentrations, seasonal variability, and ice nucleation mechanisms. Several aerosol types have been identified as potential INP sources, including black carbon, bacteria, pollen, fungal spores, and marine aerosols, but mineral dust stands out as the most relevant at the global scale (Vergara-Temprado et al., 2018b; Daily et al., 2022; Chatziparaschos et al., 2025). Within mineral dust, K-feldspar (and to a lesser extent quartz) have been shown in laboratory studies to be particularly efficient INPs (Murray et al., 2012; Kanji et al., 2017; Atkinson

et al., 2013). These findings have inspired the development of INP parameterizations that link ice nucleation efficiency to aerosol composition and size distribution.

Many global climate models, including existing versions of EC-Earth3-AerChem (van Noije et al., 2021), account for ACI only through cloud condensation nuclei (CCN), omitting INPs. Recent work evaluating INP concentrations in the TM4-ECPL atmospheric chemistry model (Chatziparaschos et al., 2023, 2025), a predecessor of the Tracer Model 5 (TM5) component

used within EC-Earth3-AerChem, has highlighted the potential to expand these capabilities toward improved representation of INPs, and thus MPCs. In this study, we build on that foundation by implementing an aerosol-sensitive heterogeneous ice nucleation scheme in EC-Earth3-AerChem, explicitly linking K-feldspar, quartz, and marine organic aerosols to primary ice formation.

Previous modelling studies (DeMott et al., 2010; Vergara-Temprado et al., 2017) have shown that these schemes improve

agreement with present-day INP observations compared to traditional temperature-only approaches, such as the widely used Meyers et al. (1992) parameterization, which tends to overestimate INP concentrations at relatively warm temperatures. Incorporating such aerosol-sensitive ice nucleation parameterizations should also allow MPCs to respond more realistically to changing aerosol loads across time and space. For instance, increases in global dust mass loading, driven by rising emissions from regions like North Africa and Asia, may have altered the Earth's energy balance since pre-industrial times (Kok et al.,

2023; Leung et al., 2025). These changes are expected to continue under future climate scenarios, yet Coupled Model Intercomparison Project Phase 6 (CMIP6) models have struggled to reproduce historical dust trends. Capturing the drivers of INP emissions along with their influence on cloud formation, is thus crucial for reliable future climate projections.

Laboratory and field studies have yielded several parameterizations suitable for integration into ESMs. These include those based on dust mineralogy (e.g., Atkinson et al., 2013; Harrison et al., 2019), marine organic aerosols (Wilson et al., 2015;



McCluskey et al., 2018b), and other INP types such as soot (Ullrich et al., 2017) and terrestrial bioaerosols (Murray et al., 2021). However, even the most comprehensive INP parameterizations fail to fully account for observed ice crystal number concentration (ICNC) in clouds, which often exceed expected values by orders of magnitude (Field et al., 2017; DeMott et al., 2010). This discrepancy highlights the potential importance of SIP processes, such as rime splintering, droplet shattering, and ice-ice collisional breakup, which can multiply ice crystals independently of primary nucleation (Pruppacher and Klett, 1997;

Field et al., 2017; Wex et al., 2019; Korolev and Leisner, 2020; Sotiropoulou et al., 2020; Järvinen et al., 2022; Georgakaki and Nenes, 2024). Despite their potential significance, SIP mechanisms are not fully understood, and some current models, including EC-Earth3, lack the necessary diagnostics for physically-based SIP schemes. Recent machine-learning approaches, such as the Random Forest Secondary Ice Production (RaFSIP) (Georgakaki and Nenes, 2024), offer promising alternatives for activating SIP in large-scale models.

In this study, we present new developments in EC-Earth3-AerChem aimed at investigating the climatic impacts of explicitly representing primary and secondary ice formation processes in MPCs. Specifically, we implement a heterogeneous ice nucleation scheme that links key ice nucleating aerosol species–K-feldspar, quartz, and marine organic aerosols–to primary ice formation, enabling the model MPCs to respond to aerosol variability. In parallel, we incorporate a machine-learning-based parameterization for SIP, allowing for more physically plausible ICNC through mechanisms such as rime splintering and col-

lisional breakup. Our focus is to assess how these enhanced process representations affect cloud properties, radiative fluxes, precipitation, and ultimately the Earth's energy balance on a global scale.

The manuscript is structured as follows: Section 2 outlines the model description–including the newly implemented developments–, the experimental setup, and the observational datasets used for model evaluation. In Section 3, we evaluate the aerosol-sensitive INP schemes by comparing the simulated INPs with in-situ INP measurements, analyze resulting ICNC distributions,

and assess the influence of both primary and secondary ice processes on cloud properties and radiative fluxes. Section 4 summarizes the key findings, discusses implications for model performance, and outlines directions for future research.

## 2 Methodology

### 2.1 Description of EC-Earth3-AerChem

The model used in this study is EC-Earth3-AerChem, the EC-Earth3 configuration with interactive chemistry and aerosols (van

Noije et al., 2021). The baseline version of this model was used to contribute to CMIP6 and AerChemMIP. This study includes several new developments: (1) explicit incorporation of the atmospheric cycle of dust minerals linked to a soil mineralogy atlas (Claquin et al., 1999), (2) online calculation of marine organic aerosol emissions, (3) aerosol-sensitive heterogeneous ice nucleation parameterizations for immersion freezing of dust minerals (K-feldspar and quartz) and marine organic aerosols in the MPC regime, and (4) a SIP parameterization based on a random forest model as detailed in the following subsections.

The atmospheric component is based on the Integrated Forecasting System (IFS, CY36R4 version) of European Centre for Medium-Range Weather Forecasts (ECMWF) at TL255L91, with a linear N128 reduced Gaussian grid corresponding to a resolution of about 80 km, and with 91 vertical levels and a model top at 0.01 hPa. It uses a timestep of 2700 s. The global



chemistry transport model TM5 (Huijnen et al., 2010) is coupled to IFS every 6 hours and operates at 3° x 2° resolution with 34 vertical levels. TM5 uses the M7 aerosol microphysics module (Vignati et al., 2004), which includes seven modes: four water-soluble (nucleation, Aitken, accumulation, and coarse), and three insoluble (Aitken, accumulation, and coarse). The aerosol species represented by M7 are mineral dust, black carbon and organic carbon, sulfate, and sea salt. In addition, TM5 simulates ammonium nitrate and methane sulfonic acid (MSA).

IFS CY36R4 includes a 1-moment cloud microphysics scheme that prognoses the mass of four classes of hydrometeors (liquid, ice, rain and snow) (Forbes et al., 2011). Ice nucleation is parameterized in a strongly simplified way, with no attempt to predict ICNC produced by nucleation, as described in Tompkins et al. (2007). At temperatures colder than -38 °C water droplets are assumed to homogeneously freeze instantaneously. Between -38 to 0 °C, the scheme allows supercooled liquid water to coexist with ice. The ice crystals then can grow at the expense of the water droplets through the Wegener-Bergeron-Findeisen process (Wegener, 1911; Bergeron, 1935; Findeisen, 1938). To compute the depositional growth rate (Pruppacher and Klett, 1997; Rotstayn et al., 2000), the ICNC must be known, under the assumption that heterogeneous ice nucleation has taken place in the supersaturated environment. However, as previously mentioned, no prognostic equation is included for ICNC. Instead, ICNC ($m^{-3}$) is determined diagnostically, following the deposition-condensation-freezing (temperature-based) nucleation parameterization of Meyers et al. (1992):

$$icnc_{Meyers} = 1000 exp[12.96(e_{sl} - e_{si})/e_{si} - 0.639] \tag{1}$$

Here, $e_{sl}$ and and $e_{si}$ are the saturation vapor pressure with respect to liquid water and ice (Pa), respectively. It is important to note that the implementation of the Meyers et al. (1992) formulation in the EC-Earth3 CMIP6 version (Döscher et al., 2022) contained an error: the denominator was incorrectly set to $e_{sl}$ instead of the correct $e_{si}$. This issue was corrected in later IFS cycles (e.g., CY47R3) and has also been fixed in the IFS CY36R4 model version used for the present study. Our tests (not shown) reveal that even though the correction of Meyers' equation showed a reduction of ICNC of roughly 1 order of magnitude in the global average, this was not affecting the formation of clouds, because the depositional growth was limited by the availability of supercooled liquid water rather than ICNC; therefore, the higher ICNC concentrations had no impact on the cloud production rates. In other words, the liquid water was probably extinguished already at lower levels of ICNC, and increasing the ICNC beyond these levels had no impact because the ice concentration could not further increase by deposition (Wegener-Bergeron-Findeisen process) since the source of the water vapor, the supercooled water, was already exhausted.

As discussed above, ICNC is determined diagnostically in the single-moment scheme, and its impact on cloud formation is limited by the availability of supercooled liquid water. Despite this limitation, it is nevertheless possible to enhance the representation of ice microphysics within this framework. In the present study, we introduce new aerosol-sensitive ice nucleation and SIP parameterizations for ICNC, which are consistently linked to the prognostic ice mass (see Sect. 2.2.2 and 2.2.3). These developments allow us to explore the sensitivity of cloud properties and radiative effects to ice number variability, while remaining fully compatible with the single-moment foundation of the host model.



## 2.2 New model developments

### 2.2.1 INP-relevant tracers: dust minerals and marine organic aerosols

Relevant minerals for ice nucleation in dust, particularly K-feldspar and quartz, have been explicitly included in the model. Their emissions are calculated using a dust emission scheme based on the approach of Tegen et al. (2002), which drives the release of mineral dust particles from arid surfaces (van Noije et al., 2021). To determine the specific emissions of K-feldspar

and quartz, we rely on the soil mineralogy atlas of Claquin et al. (1999), as implemented in Myriokefalitakis et al. (2022). This atlas provides information on the mineral fractions present in the soil, which TM5 uses to estimate the composition of emitted dust, including feldspar and quartz among other minerals.

Since the size distribution of minerals in soil differs from that of airborne aerosols, we apply the Brittle Fragmentation Theory proposed by Kok (2011) to better represent the emitted particle size distribution (PSD) of each mineral. This theory

accounts for the shattering behavior of soil aggregates upon wind erosion, leading to emitted mineral size distributions that align more closely with atmospheric observations compared to approaches based solely on soil size fractions (e.g., Perlwitz et al. (2015)).

In the model, K-feldspar and quartz are traced as insoluble components in both the accumulation and coarse modes, following the methodology of Myriokefalitakis et al. (2010, 2015, 2016) and Chatziparaschos et al. (2023). These tracers serve as the

basis for implementing the new heterogeneous ice nucleation parameterization. We note that the soil mineralogy atlas provides information on total feldspar. We assume that K-feldspar represents a 35% of the total feldspar abundance (Atkinson et al., 2013). Both accumulation and coarse mode insoluble fractions are treated as externally mixed in the model.

Another key enhancement involves the integration of a new aerosol source relevant for INP: the online calculation of marine organic aerosol emissions. Marine organic aerosols are a potentially relevant source of INP, particularly in the Southern Ocean

region (DeMott et al., 2016; McCluskey et al., 2018a; Järvinen et al., 2022; Chatziparaschos et al., 2025). Their emissions are computed online within the model by simulating the partitioning between insoluble marine organics and sea salt, following the methodology detailed in Myriokefalitakis et al. (2010) and Chatziparaschos et al. (2025). These emissions are calculated as a fraction of the emitted sea-salt aerosol in the accumulation mode, with the fraction depending on the amount of chlorophyll-a (Chl-a) present in the ocean surface layer (O'Dowd et al., 2008; Vignati et al., 2010). The emitted marine organic aerosol

particles are assumed to be entirely insoluble and therefore ice-active but internally mixed with sea salt, consistent with the findings of O'Dowd et al. (2008). The sea-salt source itself is driven by wind speed using the scheme of Gong (2003), includes a temperature dependence as described in van Noije et al. (2021), and is fitted for accumulation and coarse modes as described in Vignati et al. (2010). Chl-a concentrations are derived from monthly averaged satellite observations provided by Moderate Resolution Imaging Spectroradiometer (MODIS), with a spatial resolution of 1 ° x 1 °.

### 2.2.2 Primary ice production schemes

This work incorporates aerosol-sensitive heterogeneous ice nucleation parameterizations in the MPC regime, specifically for the immersion freezing of dust minerals (K-feldspar and quartz) and marine organic aerosols. Immersion freezing, wherein



INPs are engulfed within supercooled droplets, is considered the dominant freezing pathway in MPCs (De Boer et al., 2010; Vergara-Temprado et al., 2017). Deposition nucleation, by contrast, typically occurs at colder temperatures and is not the focus here.

K-feldspar is among the most efficient atmospheric INPs in immersion freezing (Harrison et al., 2019, Fig. 8), while quartz, though less efficient, is more abundant in the atmosphere (Harrison et al., 2019; Chatziparaschos et al., 2023). Soot particles have been excluded due to their comparatively limited role in immersion freezing (Ullrich et al., 2017). Marine organic aerosols, although considered of secondary importance on a global scale, have been shown to play a significant role on a regional scale in certain temperature regimes (Gantt and Meskhidze, 2013; Vergara-Temprado et al., 2018a; McCluskey et al., 2018b; Zhao et al.,

2021), particularly over oceanic regions like the Southern Ocean. We decided to neglect terrestrial bioaerosols, such as pollen, bacteria, and fungal spores, as INPs. In Chatziparaschos et al. (2025), better agreement with observed INP concentrations was achieved when only mineral dust and marine organic aerosols were included, while the inclusion of terrestrial bioaerosols tended to overestimate INP levels. The underlying cause of this overestimation remains uncertain.

Harrison et al. (2019) provide active site density parameterizations derived from laboratory analysis of freshly milled quartz, K-feldspar, plagioclase and albite, the former two being the most ice-active. These are implemented as:

$$icnc_{Harrison} = \sum_i [n_{Kf,i}(1 - \exp[-n_{s\text{-}Kf}\,\sigma_i]) + n_{q,i}(1 - \exp[-n_{s\text{-}q}\,\sigma_i])] \tag{2}$$

Where $n_{Kf,i}$ and $n_{q,i}$ are the number concentrations of K-feldspar and quartz particles (m$^{-3}$) in mode $i$ (i.e., accumulation and coarse modes), $\sigma_i$ the particles' surface area (cm$^2$) (computed assuming externally mixed particles) in mode i, and

$n_{s\text{-}Kf}$ and $n_{s\text{-}q}$ the time- and mode-independent nucleation site density per unit surface area (cm$^{-2}$) of K-feldspar and quartz, respectively, defined as:

$$n_{s\text{-}Kf} = 10^{[-3.25 - 0.793T - 6.91\times10^{-2}T^2 - 4.17\times10^{-3}T^3 - 1.05\times10^{-4}T^4 - 9.08\times10^{-7}T^5]} \tag{3}$$

$$n_{s\text{-}q} = 10^{[-1.709 + 2.66\times10^{-4}T^3 + 1.75\times10^{-2}T^2 + 7\times10^{-2}T]} \tag{4}$$

Here, $T$ is the temperature in degrees Celsius. Eq. 3 for K-feldspar is valid from -37.5 to -3.5 °C and Eq. 4 for quartz is valid

from -37.5 to -10.5 °C.

With this formulation, K-feldspar demonstrates high nucleation site densities ($10^6$–$10^9$ cm$^{-2}$) at cold temperatures (-37.5 to -24 °C), and remains more efficient than quartz at intermediate temperatures (-24 to -10 °C). Only K-feldspar contributes significantly at warmer temperatures (-10 to -3.5 °C). These findings align with previous studies, such as Atkinson et al. (2013), who also observed high K-feldspar efficiencies within a more restricted temperature range (-25.15 to -5.15 °C). Despite the

lower efficiency at warmer temperatures (-15 to -5.15 °C) in Atkinson et al. (2013), K-feldspar remains a crucial INP in supercooled liquid environments, reinforcing its role in MPC formation.



Marine organic aerosol immersion freezing is estimated using the parameterization by Wilson et al. (2015):

$$icnc_{Wilson} = TOC \times exp[11.2186 - (0.4459T)] \tag{5}$$

Where $TOC$ is the total organic carbon (g C m$^{-3}$) and $T$ is the temperature in degrees Celsius. Eq. 5 is valid between -27
to -6.5 °C. $TOC$ is derived following the approach of McCluskey et al. (2018b) using:

$$TOC = \frac{TOM}{OM:OC} \tag{6}$$

Where $TOM$ is the total organic matter mass concentration of marine organic aerosols in the accumulation mode and
$OM:OC$ is the ratio of organic matter to organic carbon, set to 1.8 (Vignati et al., 2010).

### 2.2.3 Secondary ice production schemes

We also incorporate SIP processes, which have the potential to enhance ice crystal concentrations under specific conditions,
particularly between -25 and 0 °C (Luke et al., 2021; Zhao and Liu, 2021; Georgakaki and Nenes, 2024).

For this purpose, we implement two versions of the RaFSIP scheme developed by Georgakaki and Nenes (2024). These
parameterizations are based on machine learning models trained using high-resolution regional climate simulations with the
Weather Research and Forecasting Model (WRF) model. RaFSIP version 1 (RaFSIPv1) estimates secondary ice crystals via
Ice Enhancement Factors (IEFs) relative to primary ice concentrations, and is active in the -20 to -3 °C temperature range. Its
limitation to -3°C arises from the requirement for primary ice to apply the IEFs, as no ICNC from primary ice is available at
warmer temperatures. RaFSIP version 2 (RaFSIPv2) predicts secondary ice crystal concentrations directly from the random
forest regressor, making it independent of primary ice and active from -20 to 0 °C. Using both versions allows us to evaluate
the sensitivity of SIP predictions to different modeling approaches–one relative to primary ice concentrations (RaFSIPv1)
and one predicting ice concentrations directly (RaFSIPv2). Both versions simulate three key ice multiplication mechanisms:
Hallett–Mossop rime-splintering (-8 to -3 °C) (Hallett and Mossop, 1974), droplet freezing and shattering (-20 to -3 °C) (Griggs
and Choularton, 1983; Phillips et al., 2018), and collisional fracturing and breakup (RaFSIPv1: -20 to -3 °C, RaFSIPv2: from
-20 to 0 °C) (Vardiman, 1978; Griggs and Choularton, 1986). Both RaFSIP versions required the following inputs from the host
model: ice water path (IWP), liquid water path (LWP), temperature, relative humidity with respect to ice (RHI), rain, liquid, ice
and snow water content (RWC, LWC, IWC, SWC, respectively), and cloud and rain riming tendencies, which represent the rate
at which supercooled droplets and raindrops freeze upon contact with ice particles, respectively. Since riming tendencies are
not diagnosed in EC-Earth3-AerChem, RaFSIP internally estimates them based on the predicted cloud and rain mass mixing
ratios at each time step (see Frostenberg et al. (2025) for details on the implementation in EC-Earth, in particular Section
A.2, which describes the random forest used to predict the riming tendencies). To prevent excessive SIP activity that could
destabilize the model, we imposed upper bounds on the SIP-induced ICNC formation rates, with RaFSIPv1 capped at 100
kg$^{-1}$s$^{-1}$, and RaFSIPv2 capped at 10 kg$^{-1}$s$^{-1}$. These thresholds were established after extensive testing to ensure numerical
stability without afecting physical realism.





The combined effect of the aerosol-sensitive primary ice parameterizations and the RaFSIP schemes is evaluated against the default temperature-based scheme by Meyers et al. (1992), which serves as the baseline for comparison (see Sect. 3).

So far, the aerosol-sensitive ice nucleation parameterization has been integrated with other developments from the EU FORCeS project to help improve the representation of aerosols and their interactions with warm and cold clouds (Thomas et al., 2024).

Finally, we note that all figures displaying ICNC in this study exclusively show ice crystals formed in the MPC regime via heterogeneous nucleation or SIP, as those are the only ice crystals contributing to depositional growth in the model.

### 245 2.3 Experimental setup

To assess the impact of the new MPC developments described in Sec. 2.2, we designed a set of targeted numerical experiments, each spanning 12 years under present-day climate conditions (2009-2020). This period was selected to align with a period with satellite observations for model evaluation.

All simulations were conducted in atmosphere-only mode using the Atmospheric Model Intercomparison Project (AMIP)
configuration, which prescribes sea surface temperature and sea ice concentrations from the Program for Climate Model Diagnosis and Intercomparison (PCMDIv1.1.8) dataset (Meehl et al., 2007).

To facilitate direct comparison with observations and to isolate the short-term cloud responses under different parameterizations, we applied a nudging technique above the PBL. Specifically, wind divergence and vorticity fields were gently nudged towards ECMWF Reanalysis v5 (ERA5) data (Hersbach et al., 2017). While nudging helps reduce biases and drifts, improving
the model's alignment with real-world conditions, it can also dampen aspects of the fast response. Nevertheless, comparisons between nudged and free-running simulations, included in the supplementary material (Figs. S1, S2 and S3), show minimal differences in zonal mean fields, supporting the robustness of our approach.

Anthropogenic and open biomass burning emissions required by the TM5 chemical transport model were taken from the CMIP6 Community Emissions Data System (CEDS; Hoesly et al. 2018) and BB4CMIP (van Marle et al., 2017) data sets for
the historical period (2009–2014) and from the SSP2-4.5 scenario for the near-present years (2015–2020) (Gidden et al., 2019). A 6-month spin-up phase (July–December 2008) preceded the main simulation period, allowing the atmospheric composition to equilibrate with newly introduced aerosols such as K-feldspar and quartz.

All diagnostics and analyses presented are based on monthly mean values averaged over the entire simulation period.

We conducted five distinct experiments, each using a different ice nucleation parameterization scheme for the MPC regime:

1. Meyers: The baseline temperature-dependent deposition-condensation-freezing parameterization by Meyers et al. (1992).

   2. Harrison: The dust-sensitive parameterization accounting for the immersion freezing of K-feldspar and quartz (Harrison et al., 2019).

   3. Harrison + Wilson (H+W): As above, with the addition of marine organic aerosol via the Wilson et al. (2015) parameterization.



4. H+W+RaFSIPv1: Incorporates the H+W primary scheme together with RaFSIPv1 (Georgakaki and Nenes, 2024), based on ice enhancement factors.

5. H+W+RaFSIPv2: Same as above, but using RaFSIPv2 (Georgakaki and Nenes, 2024), which predicts secondary ice directly via the random forest model, independently of primary ice.

These experiments allow us to systematically evaluate the individual and combined effects of aerosol-sensitive primary and secondary ice nucleation processes on cloud properties, ice crystal concentrations, and radiation fields.

### 2.4 Observational data for model evaluation

The model has been evaluated with an extensive observational dataset of in-situ INP concentrations as in Chatziparaschos et al. (2025), including the "Impact of Biogenic versus Anthropogenic emissions on Clouds and Climate: towards a Holistic UnderStanding" (BACCHUS) database (https://www.bacchus-env.eu/in/search.php, last access: 1 July 2025) with globally distributed measurements from several campaigns: Wex et al. (2019), with observations of dust-INPs in the Arctic; Tatzelt et al. (2020) and McCluskey et al. (2018a), reporting INPs over the Southern Ocean; and Welti et al. (2020), presenting ship-based measurements of INP concentrations over the Arctic, Atlantic, Pacific, and Southern oceans. See Fig. S4 and Chatziparaschos et al. (2025) for all the datasets and location details. For each observation, the modeled INP concentration is calculated at the instrument's temperature, rather than the model's own temperature, to ensure consistency in the quantitative evaluation of the ice nucleating schemes given the strong temperature dependence of INP parameterizations.

Cloud-related variables simulated by the model have been evaluated against the MODIS Cloud Feedback Model Inter-comparison Project Observation Simulator Package (COSP)-ready Cloud Properties including both Aqua and Terra satellites (https://ladsweb.modaps.eosdis.nasa.gov/missions-and-measurements/products/MCD06COSP_M3_MODIS, last access: 30 August 2023). This product is specifically designed to facilitate comparisons with climate model output using the COSP simulator (Bodas-Salcedo et al., 2011). Specifically, the COSP simulator transforms climate model output into synthetic satellite observations, enabling a comparison with MODIS retrievals while accounting for the limitations and biases inherent in satellite measurements. The simulator uses the model's 3D cloud fields, applying cloud overlap assumptions, radiative transfer calculations, and sensor geometry effects to produce variables comparable to satellite products (Bodas-Salcedo et al., 2011). Variables evaluated include cloud mask fraction (or cloud cover), IWP, and LWP. Note that both MODIS retrievals and COSP MODIS products represent in-cloud means, rather than grid cell averages. Therefore, the LWP and IWP variables were scaled using the corresponding instantaneous liquid and ice cloud fractions, respectively, to ensure consistency with the global model outputs (Pincus et al., 2012, 2023). Because retrievals in polar regions are subject to increased uncertainty (Khanal and Wang, 2018), we limited our MODIS-based evaluation to the latitudinal band from 60°S to 60°N.

For vertical cloud structure and cloud fraction evaluation, we used the Cloud-Aerosol Lidar and Infrared Pathfinder Satellite Observation (CALIPSO) General Circulation Models (GCM) Oriented Cloud CALIPSO Product (GOCCP) dataset (https://climserv.ipsl.polytechnique.fr/cfmip-obs/Calipso_goccp.html, last access: 30 August 2023), processed by Laboratory of Dynamic Meteorology of the Pierre-Simon Laplace Institute (LMD/IPSL) with support from National Aeronautics and Space



Administration (NASA) and National Centre for Space Studies (CNES). CALIPSO-GOCCP combines lidar-based vertical cloud profiling with passive sensors and is designed to evaluate cloud diagnostics in GCMs using COSP. It provides 2D fields

of low, middle, and high cloud fraction, directly comparable to the COSP-CALIPSO output. Due to its sun-synchronous near-polar orbit, CALIPSO does not provide data poleward of approximately 82°N and 82°S, limiting our evaluation domain to these latitudinal bounds.

For radiative fluxes, we used the Clouds and the Earth's Radiant Energy System (CERES) Energy Balanced and Filled (EBAF) Ed4.2 Level-3b dataset (Loeb et al., 2018; Kato et al., 2018) to evaluate top-of-atmosphere (TOA) and surface SW

and LW CRE. These data cover the full simulation period (2009–2020) and provide high-quality, observation-based estimates of Earth's radiation budget.

## 3 Results and discussion

### 3.1 Primary ice nucleation: global INP distributions and evaluation

The choice of primary ice nucleation (PIN) parameterization exerts a strong control on the spatial distribution of ice nucleating

particles (INPs), and thus ICNC, in the model. Figure 1 compares the annual mean column INP load and zonal-mean vertical distributions produced by three schemes–a purely temperature-dependent scheme (Meyers), an aerosol-sensitive dust scheme (Harrison), and a marine organic aerosol-sensitive scheme (Wilson)–as well as their combination (H+W). Distinct differences emerge among these PIN schemes (Sect. 2.2.2).

The Meyers scheme produces high INP concentration across all regions and altitudes (Fig. 1a,b). By contrast, the dust-based

scheme of Harrison yields elevated INP concentrations primarily at cold upper levels and near major desert source regions and downwind transport pathways (Fig. 1d-e). The marine organic aerosol immersion freezing scheme of Wilson produces INP concentrations several orders of magnitude lower than the other schemes (Fig. 1g-h), consistent with prior global estimates (Wilson et al., 2015; Vergara-Temprado et al., 2017).

When both aerosol species are considered together (H+W), the INP burden increases modestly compared to Harrison scheme,

primarily at high latitudes, with visible effects in the SH lowest altitudes (Fig. 1j-k). All in all, the original temperature-based scheme of Meyers produces substantially more ice crystal burden (Fig. 1a,j), indicating that aerosol limitation in H+W leads to fewer primary ice crystals aloft (Fig. 1b,k).

Among the tested schemes, the combination of dust minerals and marine organic aerosol parameterizations shows the best agreement with INP observations (Fig. 1c, f, i and l). While the correlation coefficients ($R_{log}$) across all schemes are similar

(ranging from 0.81 to 0.86), the fractions of modeled INP values within 1 ($Pt_1$) and 1.5 ($P_{t1.5}$) orders of magnitude differ notably. The Meyers scheme overestimates and the Harrison scheme, which only accounts for dust, underestimates the observed INP concentrations at the warmest temperatures (Fig. 1c and f). H+W achives the best scores (Fig. 1l). Specifically, over 50% of modeled values fall within 1 order of magnitude, and 74.5% within 1.5 orders. These results are consistent with the evaluation scores shown in Fig. 2 of McCluskey et al. (2019), where the marine organic aerosol parameterizations also showed strong

agreement with in situ observations.



In summary, this combined aerosol-sensitive setup captures a more realistic spatial distribution of INPs, reflecting both major continental dust sources and widespread oceanic sources, as well as known long-range transport patterns. Indeed, the H+W configuration aligns with earlier studies that anticipated enhanced INPs over the Southern Ocean when accounting for marine organics (McCluskey et al., 2018b; Chatziparaschos et al., 2023) and provides the best agreement with observations.

It should be noted that most available INP observations are taken near the surface; thus, this evaluation primarily reflects the model's ability to reproduce INP concentrations close to the surface. While the combined aerosol-sensitive scheme performs well at the surface, it does not necessarily imply that marine organics dominate ice formation aloft in MPCs. In higher-altitude cloud regions (where observations are sparse), mineral dust may remain the primary INP source – a caveat to bear in mind when interpreting model skill and impacts upon clouds.

### 345    3.2    Secondary ice production and ice crystal number concentrations

We now assess how adding SIP processes further influences ICNC relative to PIN at relevant MPC altitudes (approximately 0 °C to –38 °C in our model). Introducing the RaFSIP parameterization yields a dramatic increase in ICNC within the MPC zone (Fig. 2). With the inclusion of the RaFSIPv2 scheme (simulation "H+W+RaFSIPv2"), ICNCs surge throughout much of the troposphere, especially at low-to-mid altitudes (roughly 1–5.5 km). This enhancement is particularly pronounced over the

Southern Ocean and other SH regions with low primary ICNC (Fig. 2e), underscoring that SIP can compensate for the dearth of primary ice nuclei in pristine regions. Fig. 2e-f quantify the SIP-induced ICNC enhancement, confirming the central role of RaFSIPv2 in boosting ICNC between 0 and -20°C in the NH and down to -30 °C in the SH.

Interestingly, at higher altitudes, where colder conditions prevail, adding RaFSIPv2 has a more subtle or even slightly decreasing effect on ICNC, particularly in the NH (Fig. 2e-f). This small reduction at cold levels likely reflects feedbacks: as

SIP greatly increases ice at lower altitudes, it can alter local temperature/humidity profiles or deplete vapor such that homogeneous/primary ice formation aloft is mildly suppressed. Such vertical redistributions are consistent with expectations, since RaFSIPv2 primarily activates in the lower MPC regime (below 5–6 km). Overall, however, the net effect of SIP (RaFSIPv2) is a large increase in column ICNC, effectively seeding many more ice crystals in regions that were ice-poor with primary nucleation alone.

The contrast between the two SIP versions implemented is also noteworthy. The RaFSIPv1 scheme exerts only a minor influence on global ICNC patterns (see Supplemental material Fig. S5). This limited effect is due to its reliance on concurrent PIN. In regions or times with very few primary ice crystals (e.g. the remote SH or lower altitudes), RaFSIPv1 remains largely inactive, rendering its impact negligible. RaFSIPv2, by design, overcomes this limitation. It can initiate secondary ice via additional pathways that do not strictly depend on co-occurring primary ice – for example, seeder–feeder processes where falling

ice from upper clouds seeds lower clouds, or ice–ice collisional breakup during riming (Hallett–Mossop-type processes) at relatively warm sub-zero temperatures. One such pathway included in RaFSIPv2 is the "forestBRwarm" mechanism (Georgakaki and Nenes, 2024), representing collisional breakup from large ice in warmer (–3 to 0 °C) zones. These enhancements make RaFSIPv2 much more effective at producing secondary ice in a variety of conditions, yielding a more realistic and ubiquitous SIP contribution (Georgakaki and Nenes, 2024).

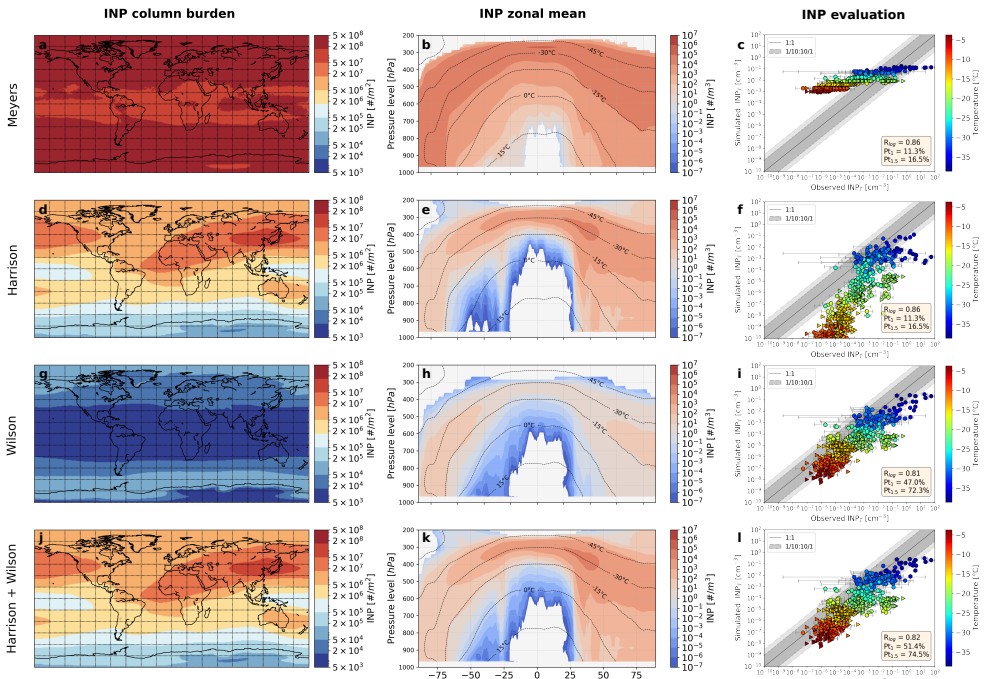

**Figure 1.** Left panels show the multi-annual (2009-2020) global distribution of column INP load when using (a) the Meyers et al. (1992) parameterization, (b) the Harrison et al. (2019) parameterization for K-feldspar and quartz minerals, (c) the Wilson et al. (2015) parameterization for marine organic aerosols (these results have been obtained them from the "Wilson" INP tracer from the H+W simulation), and (d) a two-species representation based on K-feldspar and quartz (Harrison et al., 2019) and marine organic aerosols (Wilson et al., 2015). Middle panels show the corresponding multi-annual zonal means of the INP concentration (isotherms represent all-sky grid-cell atmospheric temperature averages, not in-cloud means). Right panels show the performance of the modelled INP concentrations from the different parameterizations tested against field measurements. The dashed lines represent one order of magnitude of difference between modelled and observed, and the dashed-dotted lines represent 1.5 orders of magnitude. $Pt_1$ and $Pt_{1.5}$ are the percentages of data points reproduced within an order of magnitude and 1.5 orders of magnitude, respectively. $R_{log}$ is the correlation coefficient, which is calculated with the logarithm of the values. Triangles correspond to measurements (Bigg, 1990, 1973; Yin et al., 2012) that are compared with the climatological monthly mean simulations. Circles indicate comparisons between temporally and spatially co-located observations and model results. The locations of the data used to create this figure are shown in Fig. S4.

In summary, the advanced SIP scheme (RaFSIPv2) is crucial for capturing SIP in the model, especially in regions where primary nucleation alone may underestimate ice crystal numbers. This sets the stage for understanding how the different ice



nucleation setups (temperature-dependent vs. aerosol-sensitive primary, with and without SIP) affect cloud properties and radiation in the next sections.

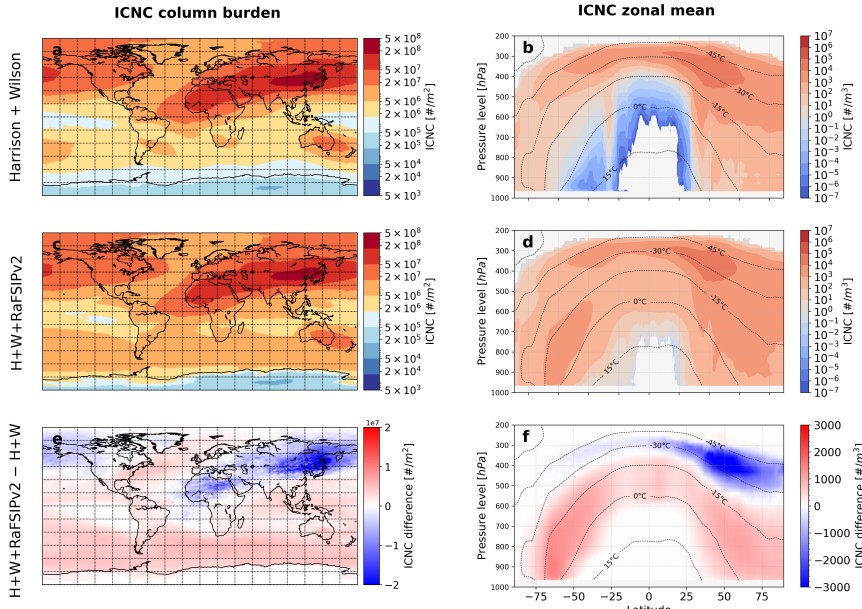

**Figure 2.** Left panels show the multi-annual (2009-2020) global distribution of column MPCs ICNC load when using (a) the Harrison et al. (2019) parameterization for K-feldspar and quartz minerals in combination with the Wilson et al. (2015) for marine organic aerosols, (b) the Harrison et al. (2019) in combination with Wilson et al. (2015) and RaFSIPv2 parameterizations, and (c) the difference between (b) and (a). The right panels show the corresponding multi-annual zonal means of the MPCs ICNC concentration (isotherms represent all-sky grid-cell atmospheric temperature averages, not in-cloud means).

### 3.3 Impacts on cloud properties and radiation

We now evaluate how these translate into changes in cloud macrophysical properties and radiative fluxes. We focus on total cloud cover and cloud liquid and ice water paths (LWP and IWP), as well as the CRE in the SW and LW parts of the spectrum. Model results from the various ice nucleation schemes are compared against satellite observations (MODIS for LWP/IWP and cloud cover; CALIPSO-GOCCP for cloud vertical distribution; CERES-EBAF for radiative fluxes) to assess not only inter-model differences but also biases relative to the real world. However, all these results should be interpreted with caution, as the
model has only been tuned using the temperature-dependent parameterization.



### 3.3.1 Cloud cover and water path changes

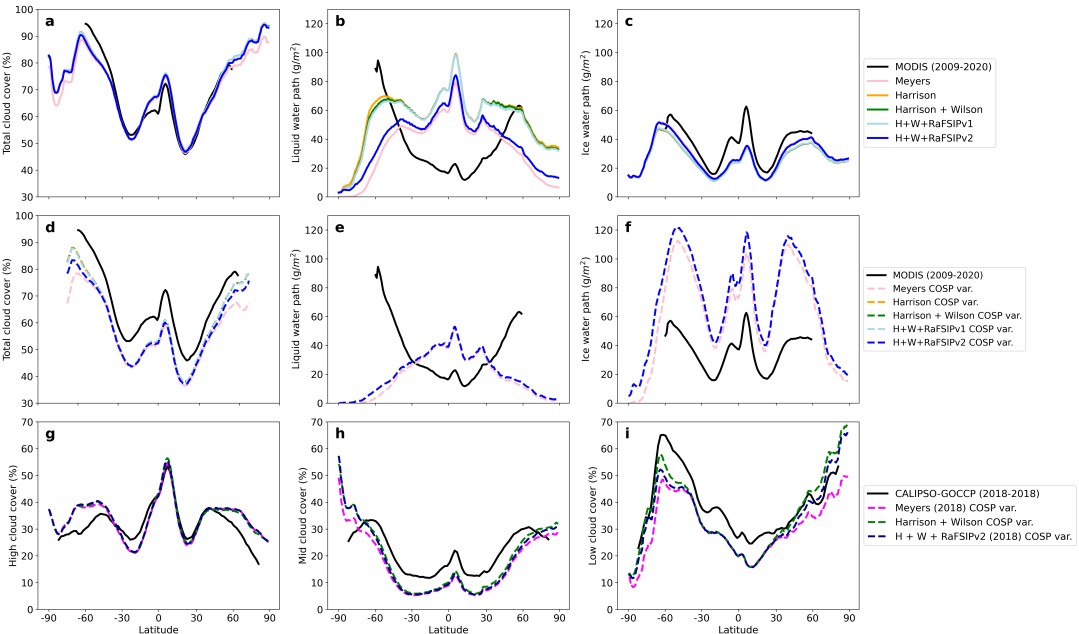

**Figure 3.** Upper row: Zonal mean of the total cloud cover, LWP, and IWP for the period 2009-2020, following Meyers et al. (1992) and the aerosol-sensitive ice nucleation parameterization by the Harrison et al. (2019), as well as its combinations with the Wilson et al. (2015), and with the Wilson et al. (2015) alongside the RaFSIPv1 and RaFSIPv2 parameterizations. Results are compared with MODIS observations. Note that in panel (c), the "Meyers" line falls below the "H+W+RaFSIPv2" and the "Harrison" and the "H+W" zonal means fall below the "H+W+RaFSIPv1" line. Mid row: Same as in the upper row but for COSP-MODIS model outputs (Unfortunately, 6-hourly outputs from the experiments "H", "H+W" and "H+W+SIPv1" were unavailable so the LWP and IWP are not shown in the e and f panels). Lower row: Zonal mean of the high, mid, and low cloud cover derived from the COSP-CALIPSO simulator for a 1-year period (2018), following Meyers et al. (1992) and the aerosol-sensitive parameterization by Harrison et al. (2019) and Wilson et al. (2015) and in combination with RaFSIPv2. Results are compared with CALIPSO-GOCCP observations.

All simulations with the new aerosol-sensitive primary nucleation (with or without SIP) show an increase in total cloud cover relative to the baseline Meyers temperature-dependent case. The differences are most apparent at high latitudes (poleward of 60°), where MPC adjustments have the largest impact on cloudiness (Fig. 3a). Specifically, H+W+RaFSIPv2 results in a global
mean cloud cover increase of 0.9 % (a relative difference of 1.4 %) compared to the Meyers setup, reaching increases of up to 12 % (mean relative difference of 3.5 %) beyond latitudes 60°N and 60°S (Fig. S6a). This reflects greater persistence of



clouds in cold regions once aerosol-sensitive primary ice and SIP are included. These additional clouds are primarily in the supercooled liquid phase (as discussed below), consistent with the model shifting toward more liquid-rich clouds when primary ice is less abundant.

The amplified cloud cover in the aerosol-sensitive setups is accompanied by changes in LWP and IWP. Compared to Meyers, the H+W+RaFSIPv2 simulation shows a notable increase in LWP – about +4.2 $\mathrm{gm}^{-2}$ globally (+9.7%) – indicating more cloud liquid water on average (Figs. 3b and S6c). Concurrently, the global mean IWP slightly decreases by roughly 0.3 $\mathrm{gm}^{-2}$ (–0.9%), meaning there is somewhat less total ice water when using the aerosol-sensitive scheme with SIP (Figs. 3c and S6d). The experiments without SIP or with RaFSIPv1 show much higher LWP and lower IWP relative to Meyers.

It is interesting that among all configurations tested, the one with full aerosol sensitivity and SIP (H+W+RaFSIPv2) actually produces cloud water amounts most akin to the Meyers simulation. In other words, although the Meyers scheme may yield excessive PIN (see Fig. 1c), and the aerosol-only H+W scheme yields high supercooled LWP (due to fewer ice crystals), the addition of SIP in H+W+RaFSIPv2 tends to remove most of these differences. This is evident in the vertical profiles of cloud condensate (Fig. 4): the H+W aerosol-sensitive scheme without SIP allows liquid water to accumulate, but activating SIP

causes some of that liquid to freeze into new ice, partially compensating for the low primary ice numbers. Thus, SIP plays a key role in regulating cloud-phase partitioning in EC-Earth-AerChem, preventing an overabundance of supercooled liquid that would otherwise occur with an aerosol-only nucleation scheme.

        H+W+RaFSIPv2 substantially increases LWC throughout much of the mixed-phase temperature range compared to Meyers (Fig. 4). The largest LWC enhancements occur around –30 °C (all-sky grid-cell average temperature), corresponding to mid-

tropospheric cloud altitudes. This indicates that suppressing primary ice (as in the aerosol-based scheme) allows more liquid to persist in clouds, especially at moderately cold temperatures. On the other hand, total IWC is generally reduced in the H+W+RaFSIPv2 run between about –15 °C and –30 °C, except in the far southern high latitudes. This reflects fewer ice crystals and less depositional growth in that key mixed-phase range when PIN is limited. However, outside of that band – at lower altitudes/warmer sub-zero temperatures and at higher altitudes/cold cloud temperatures – the aerosol-sensitive scheme

shows a slight increase in IWC relative to Meyers, likely due to adjustments.

        Including RaFSIPv2 notably alters these vertical profiles (Fig. 4d and h). SIP causes an LWC decrease in the warmer sub-zero range (0 °C down to roughly –5 to –10 °C), relative to the H+W run. In almost exactly these same regions, IWC increases with SIP. The SIP mechanisms efficiently generate ice and deplete some supercooled liquid at those levels, bringing the cloud phase mix more in line with what one would expect when ice multiplication processes operate. In summary, compared to using

aerosol-based primary ice alone, adding SIP redistributes cloud water: it draws down LWC at lower altitudes and boosts the ice content there, even as the overall cloud cover and LWP remain elevated compared to the original temperature-dependent scheme.

        Fig. 3a-f compares the model's zonal-mean cloud cover, LWP and IWP against MODIS observations, including both the direct model outputs and their COSP-simulated equivalents. The direct model output captures the meridional pattern of total

cloud cover reasonably well (Fig. 3a). Between 60°S and 60°N, the simulated cloud fraction closely matches MODIS, apart

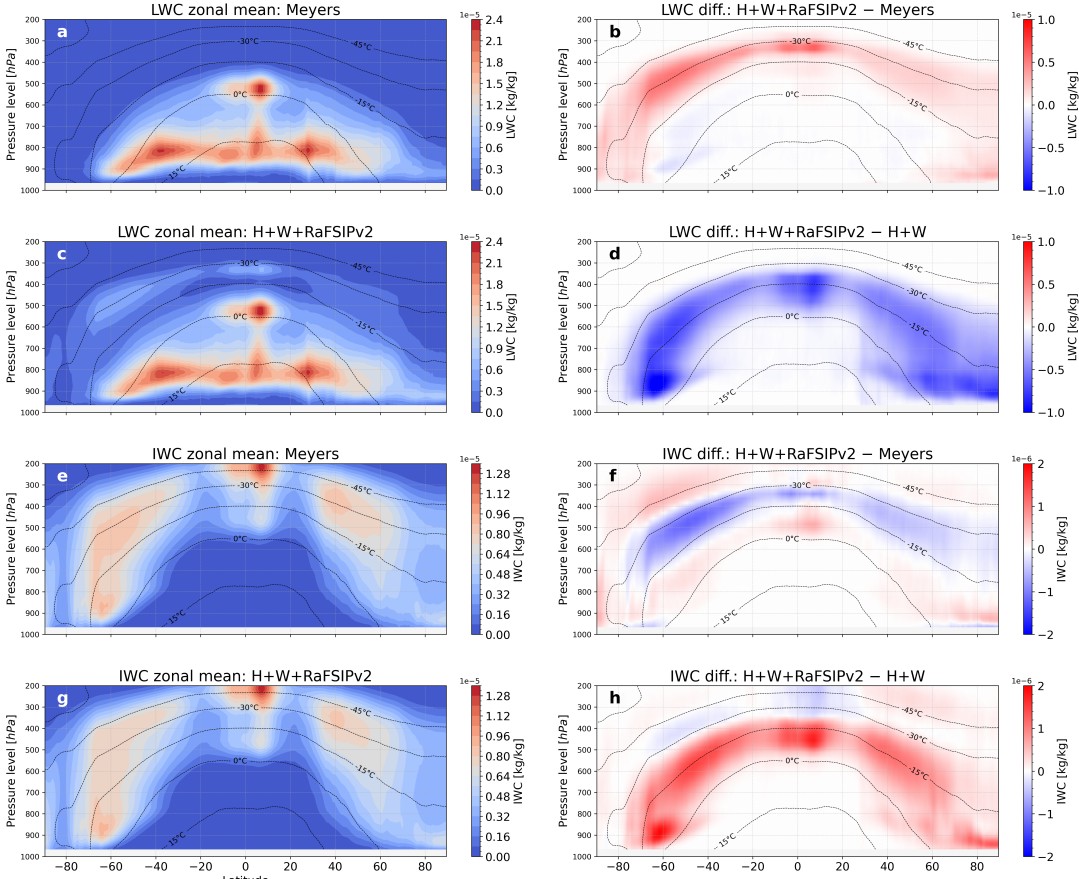

**Figure 4.** Zonal mean cross-sections of LWC and IWC for 12 years (2009–2020) from simulations using (a, e) Meyers et al. (1992) ("Meyers"), and (c, g) the aerosol-sensitive ice nucleation parameterization of Harrison et al. (2019) and Wilson et al. (2015) together with RaFSIPv2 ("H + W + RaFSIPv2"). Panels (b, f) depict the difference in LWC and IWC between the aerosol-sensitive simulation (H + W + RaFSIPv2) and the Meyers simulation, while panels (d, h) illustrate the change between the aerosol-sensitive simulation with RaFSIPv2 and the one without it (isotherms represent all-sky grid-cell atmospheric temperature averages, not in-cloud means).

from a slight underestimation in the SH and a modest overestimation in the deep tropics. In contrast, the model COSP output underestimates the MODIS observations by about 10% across all latitudes (Fig. 3d).



All model variants exhibit systematic biases in water paths. In comparison to MODIS, the direct model output tends to over-predict LWP in low-to-mid latitudes ($\approx 40°$S–$40°$N) and underpredict LWP at higher latitudes (Fig. 3b). When comparing the

COSP output, the bias improves in low-to-mid latitudes and worsens at higher latitudes. This bias pattern–too much liquid water in warm cloud regions, too little in cold cloud regions–persists regardless of the ice nucleation scheme. Meanwhile, the direct model IWP output is biased low across all latitudes (Fig. 3c), yet the COSP output significantly overestimates MODIS IWP retrievals. The apparent contradiction–a slight deficit of IWP in the direct model fields compared to MODIS but a large surplus once the same columns are processed through the MODIS simulator–may not necessarily arise from an overestimation of the

real IWP but from the way COSP translates model microphysics into satellite-like retrievals. The direct IWP output reflects the true grid-box ice mass. COSP, however, first assembles homogeneous Monte Carlo Independent Column Approximation (McICA) sub-columns that vertically stack neighbouring thin ice layers and then applies the single-layer MODIS retrieval that assumes MODIS ice optical properties. The stacked layers together with a potentially small effective ice crystal size provided by the model could yield visible optical depths that are larger than those MODIS would observe, while super-cooled mixed-

phase layers may be classified as ice by the simulator. To reconcile these unexpectedly high optical depths with the MODIS Look-Up Table (LUT) optics, the retrieval may be inflating the per-pixel IWP. Thus the COSP overestimate could be either explained by an excess ice mass, by an optical-geometry artifact, or a combination thereof.

A detailed investigation of these model biases is beyond the scope of this study. The relevant outcome of this comparison is that the main structural biases are not removed by representing PIN and SIP. Indeed, the divergences among the simulations

are smaller than their common deviations from the observations. This indicates that while our new parameterizations modify cloud properties, they do not rectify longstanding biases in EC-Earth3-AerChem's cloud simulation. Several factors contribute to these biases. First, there are known deficiencies in the model's atmospheric physics, independent of our ice nucleation schemes, which affect cloud phase partitioning and coverage. For example, the IFS (CY36R4) cloud scheme used in EC-Earth3-AerChem has an inconsistency in how supersaturation is adjusted in MPCs: it effectively treats supersaturation as zero

at 0 °C instead of extending the temperatures where supersaturation can occur to –38 °C. This quirk leads to an artificial lack of liquid water between 0 °C and –38 °C, impacting cloud cover and water content in that temperature range. Later model cycles corrected this flaw, but in our version it likely suppresses supercooled liquid and cloud fraction at mid–high latitudes. A sensitivity test (Fig. S7) confirms that fixing the supersaturation adjustment would increase cloud cover (especially of 60° in both hemispheres) and LWP (especially poleward of 30° in both hemispheres), while reducing IWP in mid-latitudes–

changes that might reduce some biases. However, we could not implement this fix without a full model retuning, since it was embedded in a broader set of code updates and altering it in isolation can destabilize the model. Secondly, the convective cloud parameterization contributes to LWP biases as it uses a fixed diagnostic temperature-dependent function by which all liquid at 0°C changes to ice at –23°C (Forbes et al., 2016). This oversimplified treatment likely contributes to the strong positive biases in the supercooled liquid fraction (SLF) at relatively warm temperatures and negative biases at colder temperatures in the

tropics. It also explains the lack of supercooled liquid water at cloud tops in mid- and high-latitude regions, especially during convective cold-air outbreaks. In later versions of the model, convection was modified to produce only liquid condensate when



cloud tops were below 600 hPa. This improvement helped reduce long-standing SW radiation biases over the Southern Ocean, North Atlantic, and North Pacific (Forbes et al., 2016).

Potential biases in MODIS retrievals should also be noted (Horváth and Davies, 2007; Minm et al., 2012; Khanal and Wang,
2018). Some findings suggest that these biases are partially attributed to the misclassification of MPCs. Khanal and Wang (2018) found that treating MPCs as liquid clouds results in LWP bias that correlate with IWP, reaching up to approximately 15% bias at an IWP of $150\,\mathrm{gm^{-2}}$ and exceeding 40% or higher when IWP is greater than $400\,\mathrm{gm^{-2}}$. Our results are in line with these findings, as discrepancies between MODIS and model simulations in liquid and ice cloud fractions (not shown) can lead to significant disparities in LWP and IWP, particularly showing large LWP and IWP biases at latitudes poleward of 35° in both
hemispheres. Another known factor contributing to high LWP bias is related to the solar zenith angle. Variations in cloud top height play a significant role in this bias due to three-dimensional radiative interactions with cloud top inhomogeneity (Khanal and Wang, 2018). Additionally, studies such as Oreopoulos and Cahalan (2005) have cautioned that MODIS retrieves only the integrated optical thickness of cloudy layers, whereas EC-Earth3-AerChem can handle inhomogeneous clouds. In regions like the Tropics, convective cumulus clouds may be obscured by thick yet relatively homogeneous cirrus anvils made of ice,
resulting in potential underestimation of LWP by MODIS. Conversely, in mid-latitudes, where storm systems may appear more inhomogeneous due to shadowing and side illumination effects, MODIS could overestimate stratiform clouds and consequently LWP (Oreopoulos and Cahalan, 2005). As shown in Duncan and Eriksson (2018), there is also a range of uncertainty associated with different sources of IWP observations and reanalyses.

All in all, our new aerosol-sensitive PIN and SIP schemes slightly improve the cloud representation (e.g., the aerosol-
inclusive run achieved a marginally lower cloud cover root mean square error (RMSE) vs. Meyers, not shown), but the broad model–observation discrepancies in cloud water paths are dominated by other model processes. Addressing those biases requires further model development beyond PIN and SIP processes as shown by more recent IFS cycles.

The vertical distribution of cloud layers was also examined using the CALIPSO-GOCCP dataset for year 2018. In this case, we used the COSP-CALIPSO simulator, following a similar approach to Boudala et al. (2022). The lower panels of Fig. 3
compare the zonal fractions of high, mid, and low cloud between the model (for the Meyers and H+W+RaFSIPv2 cases) and CALIPSO. Regardless of the parameterizations used, the model underestimates mid-level and low-level cloud cover, combined with an excess of high cloud cover at mid- to high latitudes. The H+W+RaFSIPv2 simulation produces more low and mid-level clouds than Meyers at mid-to high latitudes, but differences are small relative to the gap with CALIPSO.

### 3.3.2 Radiative fluxes and cloud radiative effects

We next evaluate how the different ice production schemes affect radiative fluxes, particularly the CRE at the TOA and at the surface. Figures 5 and 6 present the zonal-mean SW, LW, and net (SW+LW) CRE from each simulation, compared against CERES-EBAF observations (2009–2020). We focus on model biases relative to CERES and the differences between simulations to isolate the impact of the new parameterizations.

The biases in TOA SW CRE (Fig. 5a-b) are tightly linked to errors in LWP (Fig. 3b). At latitudes poleward of 40° in
both hemispheres, the model underestimates LWP, which results in clouds that are too optically thin, reflecting too little solar



radiation (i.e., a positive TOA SW CRE bias). This bias is most pronounced over the high-latitude Southern Ocean, where too little LWP and low cloud cover (Fig. 3a) lead to an overly low cloud albedo and excessive SW radiation reaching the surface. Here, H+W+RaFSIPv2 partially alleviates this high-latitude bias by increasing LWP and cloud reflectivity compared to the Meyers baseline. In contrast, in the tropics and subtropics ($\approx$40°S-40°N), the model overestimates LWP (Fig. 3b), producing

optically thick clouds that reflect too much sunlight. Consequently, the tropical TOA SW CRE is too negative (stronger cooling effect than observed), and this negative bias becomes slightly worse with the aerosol-sensitive scheme (due to its tendency to further increase liquid content in MPCs).

TOA LW CRE biases are generally smaller in magnitude than their SW counterparts, with the Meyers simulation showing the closest agreement with observations (Fig. 5d-e). Biases in LWP/IWP will not spoil excessively the TOA LW CRE so long

as cloud-top height, fractional high-cloud area, and the overlap formulation together keep the optical depth in the saturated range and distribute clear-sky gaps realistically. In comparison to Meyers, the aerosol-sensitive schemes tend to overestimate the TOA LW CRE, particularly in the higher latitudes, due to an overall increase in LW trapping by increased LWP, despite potential cloud-top temperature rises due to slight ice removal.

The magnitude of TOA LW CRE errors remains smaller than those for SW, making SW the dominant contributor to

net TOA CRE biases (Fig. 5h). Remarkably, the globally averaged RMSE of the TOA net (SW+LW) CRE obtained with H+W+RaFSIPv2 is similar to that of Meyers (10.2 and 10.3 W/m$^2$, respectively).

The isolated impact of SIP (both with RaFSIPv1 and RaFSIPv2) is shown in the third column of Fig. 5. Including RaFSIPv1 has an insignificant effect on the cloud properties and therefore has little impact on the CRE. In contrast, including RaFSIPv2 results in a net warming of the system mainly due to a decrease in LWP, as indicated by a positive net CRE at the TOA (Fig. 5i).

This net warming is particularly pronounced at high southern latitudes ($\sim$60°S). At these latitudes, the shortwave TOA CRE increases by approximately 13 W/m$^2$, while the longwave TOA CRE decreases by about 3 W/m$^2$, yielding a net TOA CRE anomaly up to +10 W/m$^2$. At the equator, the net effect is about +5 W/m$^2$, and across the mid-latitudes in the NH (30–60°N), the net TOA CRE increase is approximately +6 W/m$^2$.

Surface SW CRE patterns largely mirror TOA behavior. The SW component (Fig. 6) shows similar spatial structures to

those at the TOA, with aerosol-sensitive simulations reducing biases at mid- and high latitudes, but increasing them slightly in the tropics in comparison to the temperature-only-sensitive simulation (Meyers). The surface LW CRE biases are minimal for all simulations at low latitudes because the clear-sky atmosphere is already very warm and humid, so clouds add only a little extra downwelling LW. The bias become negative at higher latitudes, with the aerosol-sensitive simulations being closer to observations than the Meyers case mainly due to increases in LWP (Fig. 3b).

The vertical distribution of LWC plays a key role in shaping the surface CRE. At high latitudes, the aerosol-sensitive simulation increases cloud cover (Fig. 3a) and LWC throughout the vertical column (Fig. 4b) compared to the temperature-sensitive configuration, leading to enhanced SW cooling and LW warming at the surface. This combination enhances surface warming at high latitudes, with regional temperature increases reaching in our atmosphere-only experiments up to +3.2°C–substantially greater than the global average (Fig. S6b). In contrast, in the tropics, LWC increases occur mainly at higher altitudes (Fig. 4b),

strengthening LW CRE at the TOA but with limited influence on the surface energy balance. The increased cloud albedo in





these regions leads to a more negative SW CRE, resulting in slight net cooling over tropical continents in the aerosol-sensitive simulation relative to the baseline (Fig. S6b).

The global RMSE of the net surface CRE is 10.8 W/m² in the aerosol-sensitive run versus 10.0 W/m² in the Meyers case. The new aerosol-sensitive parameterization reduces the biases at mid- and high latitudes but slightly increases them at low 530 latitudes.

## 4 Conclusions and perspective

This study presents a substantial enhancement of the EC-Earth3-AerChem ESM through the implementation of an aerosol-sensitive heterogeneous ice nucleation parameterization for immersion freezing, together with a machine-learning-based SIP scheme. These improvements allow the model to explicitly account for the role of aerosol type and concentration in primary 535 ice formation and to more realistically represent the multiplication of ice crystals in mixed-phase clouds. We note, however, that the sensitivities and results obtained with EC-Earth3 may differ in other models, regardless of whether they employ single- or multi-moment microphysics, due to differences in microphysical formulations, model complexities, and parameterizations (Frostenberg et al., 2025). Testing these processes in other Earth System Models is therefore necessary to assess the robustness and generality of our findings.

A number of significant enhancements in the representation of aerosol and cloud processes have been implemented in the EC-Earth3-AerChem model. This model version explicitly incorporates the atmospheric cycles of various dust minerals, including quartz and K-feldspar, as well as marine organic aerosol particles. These new aerosol sources have allowed the implementation of several heterogeneous ice nucleation parameterizations. Alongside the implementation of random forest SIP parameterizations, we tested several parameterizations and identified a suitable alternative to the temperature-dependent 545 approximation of Meyers et al. (1992): an aerosol-sensitive parameterization that includes SIP. Our results suggest that the combination of Harrison et al. (2019), Wilson et al. (2015), and Georgakaki and Nenes (2024) (RaFSIPv2) produces a realistic ICNC field, closely linked to aerosol sources and transport regions, while performing similarly to the temperature-dependent approach.

The aerosol-sensitive scheme leads to improved agreement with global in-situ observations of ice nucleating particles (INPs), 550 particularly when K-feldspar, quartz and marine organics are included. The resulting spatial distribution of INPs is more realistic than that produced by the traditional temperature-only scheme, especially over remote oceans and regions with minimal dust input. Adding the SIP scheme—especially the more advanced RaFSIPv2—amplifies ICNC in regions with low primary INPs, such as the Southern Ocean, helping to rebalance the cloud phase.

In terms of cloud macrophysical properties, the new parameterizations increase cloud cover and LWP in mid- and high-555 latitude regions, reflecting greater persistence of supercooled liquid water. However, when SIP is not accounted for, the reduction in primary ice may lead to unrealistically high LWP, underscoring the necessity of SIP for restoring more realistic cloud phase partitioning. Comparisons with MODIS and CALIPSO observations reveal persistent structural biases in LWP and IWP, particularly in regions dominated by MPC regimes. LWP remains overestimated in the tropics and underestimated



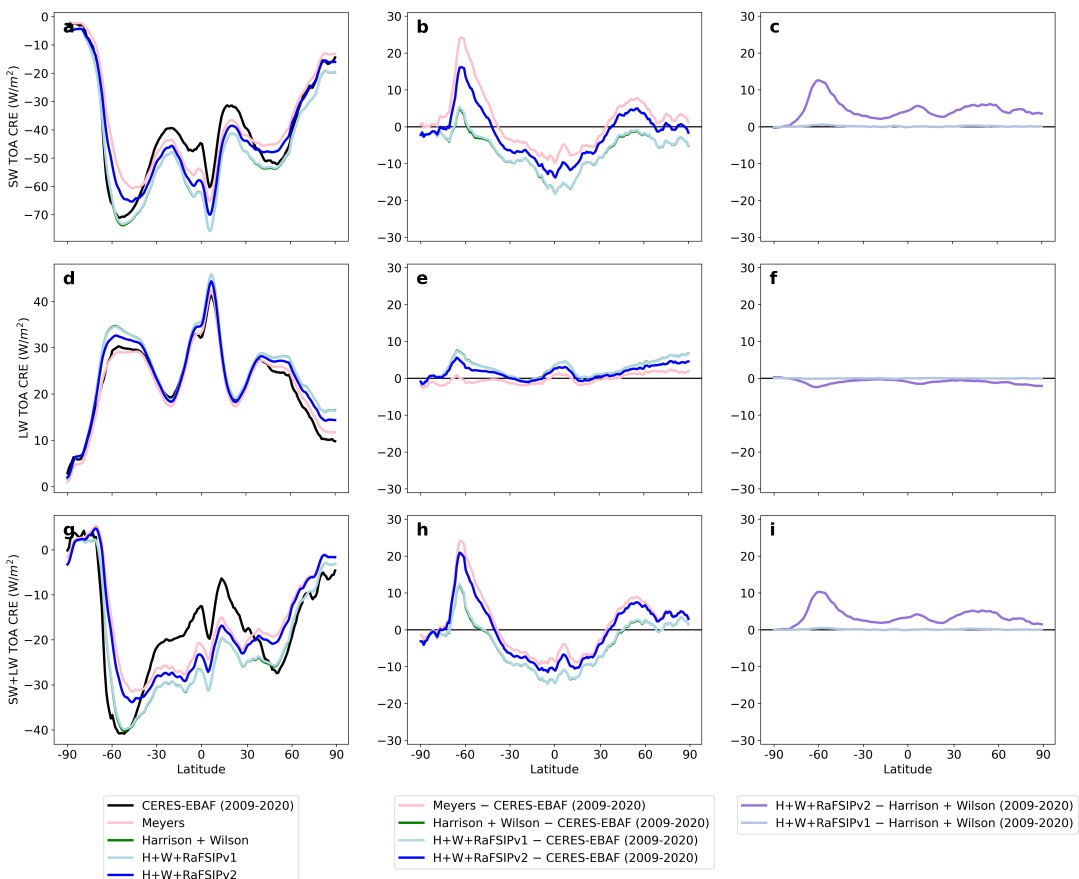

**Figure 5.** TOA CRE. Left column: Zonally averaged, climatological (2009–2020) mean TOA SW, LW and net (SW+LW) CRE, following Meyers et al. (1992), the aerosol-sensitive ice nucleation parameterization of Harrison et al. (2019) together with Wilson et al. (2015), and their combination with the RaFSIPv1 and RaFSIPv2 parameterizations. Middle column: Differences between the nudged simulations and the observations of CERES-EBAF. Right column: Radiative flux differences for the aerosol-sensitive simulation "H+W+RaFSIPv2" compared to Meyers et al. (1992), "H+W+RaFSIPv2" compared to "H+W", and "H+W+RaFSIPv1" compared to "H+W". Note that all these figures have to be read taking into account that the model has only been tuned for the ice nucleation parameterization that follows Meyers et al. (1992) (refer to Sect. 3 in van Noije et al. (2021) for more details).

at high latitudes, while COSP-simulated IWP shows discrepancies that reflect both model limitations and known challenges in
satellite retrievals.

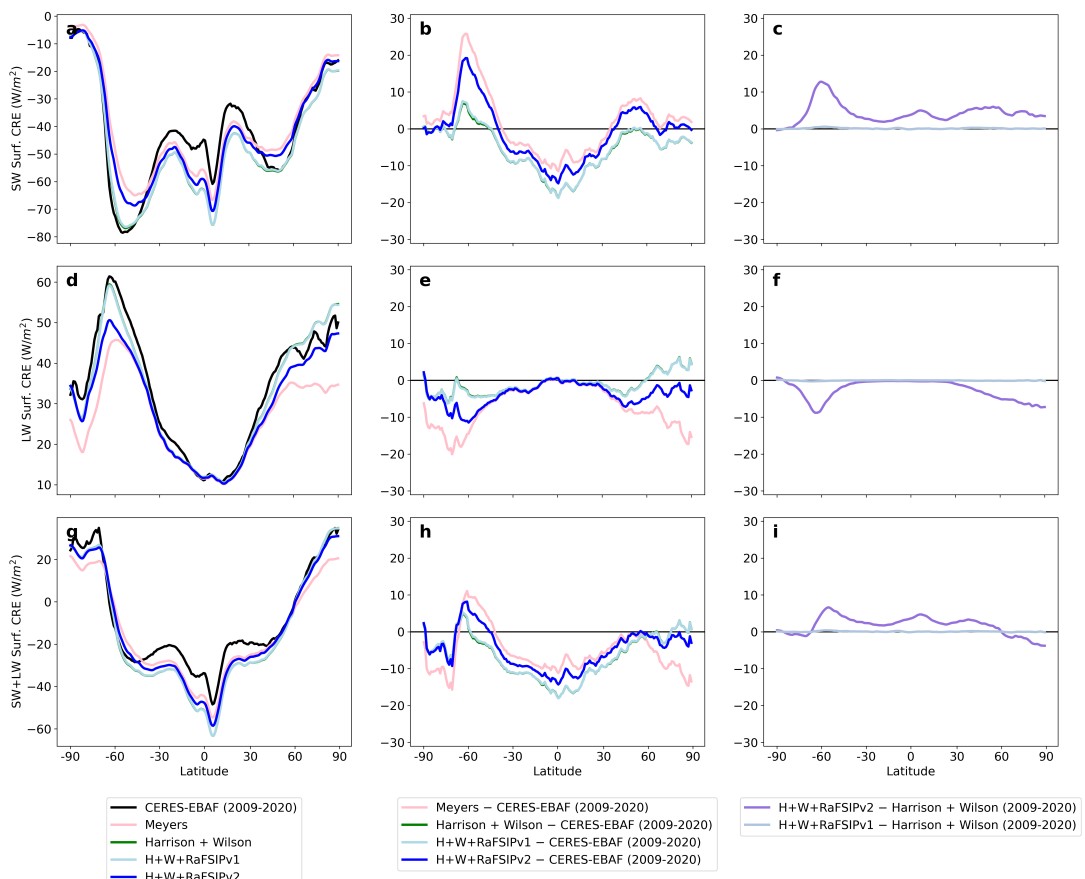

**Figure 6.** Surface CRE. Left column: Zonally averaged, climatological (2009–2020) mean surface SW, LW and net (SW+LW) CRE, following Meyers et al. (1992), the aerosol-sensitive ice nucleation parameterization of Harrison et al. (2019) together with Wilson et al. (2015), and their combination with the RaFSIPv1 and RaFSIPv2 parameterizations. Middle column: Differences between the nudged simulations and the observations of CERES-EBAF. Right column: Radiative flux differences for the aerosol-sensitive simulation "H+W+RaFSIPv2" compared to Meyers et al. (1992), "H+W+RaFSIPv2" compared to "H+W", and "H+W+RaFSIPv1" compared to "H+W". Note that all these figures have to be read taking into account that the model has only been tuned for the ice nucleation parameterization that follows Meyers et al. (1992) (refer to Sect. 3 in van Noije et al. (2021) for more details).

Evaluation of radiative fluxes shows that the aerosol-sensitive parameterization with SIP improves SW and LW CRE at mid- and high latitudes, although biases increase in the tropics. Despite these regionally different responses, the RMSE in



TOA net CRE remains essentially unchanged compared to the temperature-based Meyers scheme, indicating that the new parameterizations alter phase partitioning and cloud properties without significantly disrupting the global radiation budget.

While the aerosol-sensitive scheme introduces physically grounded improvements and allows for more nuanced diagnostics of aerosol–cloud interactions, it does not resolve all known cloud and radiation biases in EC-Earth3. Many of these shortcomings appear to originate from the structural formulation of cloud and convection parameterizations in IFS Cy36R4, for example, the hard-coded thresholds for phase transitions and simplified detrainment assumptions–rather than from the treatment of ice nucleation itself. Nevertheless, the new implementation shows clear potential to reduce certain biases, such as the cold bias

over high-latitude continental regions, and provides a framework for future work on aerosol–cloud–radiation coupling in global models.

    Current observational cloud products, such as MODIS and CALIPSO-GOCCP, greatly enhance model evaluation of cloud properties. We have found that all ice nucleation parameterizations–both temperature-dependent and aerosol-sensitive–clearly overestimate LWP at latitudes up to 40º, while large underestimations occur at higher latitudes. Yet, further efforts are required

to reduce uncertainties in retrievals, allowing researchers to confidently use these products with minimal adjustments. Additionally, improvements to the COSP simulator are crucial for better aligning models with observations (e.g., COSP2). Ongoing work to assess the accuracy and consistency of observational datasets is instrumental in advancing studies like this, ultimately enhancing their reliability and ease of integration into modeling frameworks.

    Regarding the evaluation against CERES-EBAF CRE fluxes at the TOA and at the surface, the new ice nucleation devel-

opments included in EC-Earth3-AerChem improve the comparison at mid and high latitudes while performing worse in low latitudes compared to the temperature-dependent simulation.

    The newly introduced aerosol-sensitive parameterization (including SIP) in EC-Earth3-AerChem could help mitigate some of the known EC-Earth3 biases (Döscher et al., 2022) because it may simulate the global ice formation more realistically. Specifically, the simulation with the aerosol-sensitive ice nucleation parameterization tends to warm the high-latitude regions

compared to the temperature-dependent approach (Fig. S6b). This could correct part of the cold bias found in previous studies over large areas of the NH land regions and the Arctic (Fan et al., 2020; van Noije et al., 2021; Döscher et al., 2022). However, over Antarctica, the new aerosol-sensitive parameterization leads to further warming on top of the warm bias identified by Döscher et al. (2022). The same authors attributed large parts of the warm bias of EC-Earth3 over the Southern Ocean and Antarctica to biases in SW CRE. Looking ahead, more comprehensive advances in EC-Earth's microphysics, cloud geometry,

and convective treatment will be necessary to fully realize the benefits of aerosol-sensitive and process-based ice nucleation schemes. The transition to EC-Earth4, which adopts updated IFS cycles and revised physical parameterizations, offers an important opportunity to integrate and further develop these capabilities. We anticipate that modifications in the cloud microphysical scheme from later IFS cycles, such as modifications in the cloud scheme and the representation of supercooled liquid water made in more recent versions of IFS, including CY45R1 (Forbes and Ahlgrimm, 2014; Forbes et al., 2016) will

substantially reduce these biases. The new generation of the EC-Earth ESM, EC-Earth4, will rely in the OIFS version 48r1, and will bring these updates to the ESM framework. Further improvements are expected from modifications introduced by the EC-Earth community. Specifically, the calculation of cloud droplet formation in updrafts has been improved by replacing the



Abdul-Razzak and Ghan (2000) aerosol activation parameterization for Morales Betancourt and Nenes (2014), documented in Thomas et al. (2024)), together with the introduction of the new ecRad radiation scheme in CY43R3 (Hogan et al., 2017), have
the potential to further reduce these biases.

Correcting biases in models is key for better climate predictions. Improvements in reducing the uncertainties in the estimation of MPC cloud fields are crucial, as it has been argued that the cloud optical depth will increase over the Southern Ocean due to warming-driven replacement of ice cloud content with liquid (Forster et al., 2021; Murray et al., 2021), which is optically thicker and enhances the negative feedback (Boucher et al., 2013). Furthermore, transitioning from parametrized to mechanistic
descriptions of ice crystal formation is imperative for ESMs to effectively capture other potential feedbacks within the climate system. Given that numerous aerosol sources are sensitive to climate dynamics, explicit representation of these processes in models is essential for anticipating feedback effects. Failure to incorporate such detailed descriptions may hinder the model's ability to accurately simulate the complex interactions and feedback mechanisms inherent in the climate system. We believe that our developments triggered improvements in the heterogeneous ice formation in EC-Earth that will continue in the future
to advance in this direction.

*Code and data availability.* The EC-Earth model is restricted to institutes that have signed a memorandum of understanding or letter of intent with the EC-Earth consortium and a software license agreement with the ECMWF. Confidential access to the code and the data used to produce the simulations described in this paper can be granted to editors and reviewers; please use the contact form at http://www.ec-earth.org/about/contact to obtain the code for simulation #1 or contact the corresponding author directly for access to the code used in the
rest of the simulations. All model simulation output data are stored at the Barcelona Supercomputing Center facilities in the Marenostrum supercomputer archive and are available upon request, subject to prior authorization. Please contact the corresponding author to obtain access. The Python scripts used to generate the figures in this paper are shared on Zenodo (https://doi.org/10.5281/zenodo.17175726).

Regarding the datasets used for evaluation purposes in this study, we provide a comprehensive list to ensure clarity and transparency. All external datasets cited in the Methodology and References sections are summarized below, with their URLs
and/or DOIs (where available) included to facilitate direct access and reproducibility:

– MODIS (Aqua/Terra) Cloud Properties Level 3 monthly, 1x1 degree grid (MCD06COSP_M3_MODIS): https://ladsweb.modaps.eosdis.nasa.gov/missions-and-measurements/products/MCD06COSP_M3_MODIS, DOI: 10.5067/MODIS/MCD06COSP_M3_MODIS.062

– GCM-Oriented CALIPSO Cloud Product (CALIPSO-GOCCP): https://climserv.ipsl.polytechnique.fr/cfmip-obs/Calipso_
goccp.html

– CERES-EBAF Ed4.2 Level-3b: https://ceres.larc.nasa.gov/data/, DOIs: 10.1175/JCLI-D-17-0208.1 and 10.1175/JCLI-D-17-0523.1

– BACCHUS: https://www.bacchus-env.eu/in/search.php



– Wex et al. (2019) database DOI: https://doi.org/10.1594/PANGAEA.899701

– Tatzelt et al. (2020) DOI: 10.5281/zenodo.4311665

– McCluskey et al. (2018a) DOI: 10.1029/2018GL079981

– Welti et al. (2020) DOI: 10.5194/acp-20-15191-2020

*Author contributions.* MC-S, CPG-P, and MGA, conceived the study with input and revisions from all authors. MC-S, CPG-P and MGA performed the analysis with contributions from the other authors. MC-S, MGA, CPG-P, GMP, TvN, and PLS, implemented the new aerosol-

sensitive ice nucleation parameterizations in the EC-Earth3-AerChem model, updated necessary revisions for this study, created critical input, performed and post-processed the simulations. PG and AN developed the RaFSIP parameterization, and MC-S, MGA, GMP, PG, CPG-P and PLS implemented it into EC-Earth3-AerChem. SM and MGA implemented the new mineral tracers into TM5. MC, CPG-P and MGA implemented the marine organic aerosol parameterization into TM5. MC-S and GMP introduced the new mineral tracers and aerosols into EC-Earth3-AerChem. MCS, MC, CPG-P and MGA performed the model evaluation with several kinds of observations and reanalysis data.

MCS, MGA and CPG-P, with input from all authors, wrote the manuscript.

*Competing interests.* Maria Kanakidou is a member of the editorial board of Atmospheric Chemistry and Physics.

*Acknowledgements.* The research leading to these results has received funding from the EU H2020 FORCeS project under grant agreement No. GA 821205, from the Horizon Europe FOCI project under grant agreement No. 101056783, from the Horizon Europe CERTAINTY (Cloud-aERosol inTeractions & their impActs IN The earth sYstem) project under grant agreement No. 101137680, and from the Horizon

Europe CleanCloud project under grant agreement No. 101137639. Views and opinions expressed are those of the authors only and do not necessarily reflect those of the European Union. Neither the European Union nor the granting authority can be held responsible for them. CPG-P, MC-S, MGA and MC acknowledge the funding from the European Research Council under the Horizon 2020 research and innovation programme through the ERC Consolidator Grant FRAGMENT (grant agreement No. 773051), the AXA Research Fund through the AXA Chair on Sand and Dust Storms at the Barcelona Supercomputing Center (BSC), and from the Spanish Ministerio de Economía y

Competitividad (grant PID2022-140365OB-I funded by MICIU/AEI/10.13039/501100011033 and by ERDF, EU). MK acknowledges support by the REINFORCE research project implemented in the framework of H.F.R.I. call "Basic research Financing (Horizontal support of all Sciences)" under the National Recovery and Resilience Plan "Greece 2.0" funded by the European Union – Next Generation EU (H.F.R.I. project no. 15155). MC-S acknowledges this work has received funding from the European Union's Horizon 2020 research and innovation programme under the Marie Skłodowska-Curie grant agreement No 754433. The authors gratefully acknowledge the supercomputing

resources at MareNostrum and the technical support provided by the Barcelona Supercomputing Center through the "Red Española de Supercomputación" (RES-AECT-2023-1-0008 and RES-AECT-2023-3-0026) and Partnership for Advanced Computing in Europe (PRACE) supercomputing networks.



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
