# Peer review of "Implementation of Primary and Secondary Ice Production in EC-Earth3-AerChem: Global Impacts and Insights"

_EGUsphere, 2025_

## Referee Comment (RC1)

**Review of "Implementation of Primary and Secondary Ice Production in EC-Earth3-AerChem: Global Impacts and Insights" by Costa-Suros et al.**

**General comments**

The study focuses on the improvement of the representation of mix-phase clouds (MPC) in the EC-Earth3-AerChem Earth System Model by replacing the ice nucleation scheme. The new scheme includes the effect of aerosols (K-feldspar, quartz and marine organic aerosols) in heterogeneous nucleation, as well as different mechanisms for secondary ice production (SIP) (Hallett-Mossop rime-splintering, droplet freezing and shattering, collisional fracturing and breakup).

The authors do a thorough analysis comparing the results of five experiments with in-situ and satellite observations. Each experiment includes a different ice nucleation parameterization for MPC.

Although the article might be dense, due to the large amount of information presented, it is well structured and the analysis are carefully planned with attention to detail, addressing the shortcomings of both the model and instruments used for the comparison and how this might impact the interpretation of the results. I recommend this article for publication after assessing the following minor technical comments.

**Technical comments**

Line 24: … challenging due to complex, sub-grid scale ... – … due to the complex ...

Line 135: … the higher ICNC concentrations ... – the higher ICNC (otherwise is concentration twice)

Line 253: ...PBL... – planetary boundary layer (since it is only mentioned once)

---

## Referee Comment (RC2)

This manuscript presents a model development effort by implementing aerosol-sensitive primary ice nucleation and machine-learning-based secondary ice production schemes into the EC-Earth3-AerChem Earth System Model. The study addresses a critical gap in representing mixed-phase clouds and their climatic impacts. The work is well-motivated, the methodology is largely sound, and the analysis is comprehensive, comparing multiple model configurations against a range of observational datasets. The findings that the combined aerosol-sensitive PIN and SIP scheme yields realistic INP and ICNC fields and modulates cloud-phase partitioning are valuable contributions to the field.

However, the manuscript requires revisions to improve clarity, justify methodological choices, and strengthen the interpretation and discussion of results. The current presentation sometimes lacks depth in explaining the physical mechanisms behind the modeled changes and in contextualizing the persistent model biases. The following specific comments aim to help the authors improve the manuscript.

The description of the nudging technique is brief. Please specify the nudging strength (e.g., relaxation time scale) and the vertical extent above the PBL where it is applied. Furthermore, while you mention that nudging "can also dampen aspects of the fast response," a more detailed discussion is needed on how nudging might specifically affect the evaluation of cloud microphysical responses to your new parameterizations, which are process-based and potentially sensitive to synoptic variability.

The choice of a 12-year simulation period (2009-2020) is justified for satellite overlap. However, for INP evaluation which uses specific campaign data, is this period representative of the climatological INP state, or could interannual variability (e.g., in dust emission or marine productivity) influence the comparison scores (e.g., P_t1)? A brief comment on this would be helpful.

You state that riming tendencies are not diagnosed and are internally estimated by RaFSIP. This is a crucial detail. The reader needs a clearer understanding of how this internal estimation works, as riming is a key driver for several SIP mechanisms. The reference to Frostenberg et al. (2025) is given, but since this is a "to be submitted" reference, the manuscript should include a more self-contained, concise explanation of the method or its core assumptions in the main text or a dedicated appendix.

**Lines 340-345:** The caveat regarding surface vs. aloft INP dominance is critical. This point should be elevated and expanded upon in the discussion (Section 4). What are the implications for your model's cloud impacts if mineral dust dominates ice nucleation aloft in many regions, while your combined scheme's superior performance is primarily driven by better surface INP representation from marine organics? Does this create a potential disconnect between INP evaluation skill and actual cloud impact skill?

**Lines 360-370:** The explanation for the limited impact of RaFSIPv1 is clear. However, the description of the "forestBRwarm" mechanism in RaFSIPv2 and other "additional pathways"

remains vague. Please provide a more concrete, physical description of the primary mechanisms that make RaFSIPv2 effective independent of concurrent PIN. For example, how does the scheme handle seeder-feeder processes conceptually?

**Lines 420-450:** The discussion of persistent LWP/IWP biases and the role of COSP simulator artifacts is excellent and honest. However, the conclusion that "the main structural biases are not removed by representing PIN and SIP" is somewhat buried. This should be a central, clearly stated takeaway in the abstract and conclusions.

**Lines 490-510:** The analysis of radiative effects is comprehensive. However, the interpretation of the "net warming" introduced by RaFSIPv2 (Fig. 5i) needs strengthening. Is this warming primarily due to LWP decrease (as stated) leading to reduced cloud albedo? Link this mechanism more explicitly to the physical processes in SIP (e.g., conversion of liquid to ice via rime splintering, potentially leading to faster precipitation and reduced cloud lifetime/albedo). Also, discuss the climatic significance of a +10 W/m² TOA CRE anomaly at high southern latitudes.

---

## Author Comment (AC1)

We provide detailed, point-by-point responses to the comments of Reviewers 1 and 2, with reviewer comments presented in black, and our responses, together with a summary of the corresponding revisions incorporated into the manuscript, presented in blue. This response refers to the manuscript: "Implementation of Primary and Secondary Ice Production in EC-Earth3-AerChem: Global Impacts and Insights", by Montserrat Costa-Surós, María Gonçalves Ageitos, Marios Chatziparaschos, Paraskevi Georgakaki, Manu Anna Thomas, Gilbert Montané Pinto, Stelios Myriokefalitakis, Twan van Noije, Philippe Le Sager, Maria Kanakidou, Athanasios Nenes, and Carlos Pérez García-Pando.

**REVIEWER 1:**

Review of "Implementation of Primary and Secondary Ice Production in ECEarth3-AerChem: Global Impacts and Insights" by Costa-Suros et al. General comments The study focuses on the improvement of the representation of mix-phase clouds (MPC) in the ECEarth3-AerChem Earth System Model by replacing the ice nucleation scheme. The new scheme includes the effect of aerosols (K-feldspar, quartz and marine organic aerosols) in heterogeneous nucleation, as well as different mechanisms for secondary ice production (SIP) (Hallett-Mossop rimesplintering, droplet freezing and shattering, collisional fracturing and breakup).

The authors do a thorough analysis comparing the results of five experiments with in-situ and satellite observations. Each experiment includes a different ice nucleation parameterization for MPC.

Although the article might be dense, due to the large amount of information presented, it is well structured and the analysis are carefully planned with attention to detail, addressing the shortcomings of both the model and instruments used for the comparison and how this might impact the interpretation of the results. I recommend this article for publication after assessing the following minor technical comments.

We thank the reviewer for the constructive review of our manuscript. We appreciate the positive feedback regarding the structure of the paper, the scope of the analysis, and the attention to detail in our comparison against observations. We are grateful for the reviewer's careful reading and for identifying several useful technical clarifications.

Technical comments

- Line 24: … challenging due to complex, sub-grid scale ... – … due to the complex …

We agree and have corrected the text accordingly.

- Line 135: … the higher ICNC concentrations ... – the higher ICNC (otherwise is concentration twice)

Thank you for pointing this out. We have revised the text to "the higher ICNC" to avoid redundancy.

- Line 253: ...PBL... – planetary boundary layer (since it is only mentioned once)

We have now introduced the full term "planetary boundary layer (PBL)" at its first occurrence.

**REVIEWER 2**

This manuscript presents a model development effort by implementing aerosol-sensitive primary ice nucleation and machine-learning-based secondary ice production schemes into the EC-Earth3-AerChem Earth System Model. The study addresses a critical gap in representing mixed-phase clouds and their climatic impacts. The work is well-motivated, the methodology is largely sound, and the analysis is comprehensive, comparing multiple model configurations against a range of observational datasets. The findings that the combined aerosol-sensitive PIN and SIP scheme yields realistic INP and ICNC fields and modulates cloud-phase partitioning are valuable contributions to the field. However, the manuscript requires revisions to improve clarity, justify methodological choices, and strengthen the interpretation and discussion of results. The current presentation sometimes lacks depth in explaining the physical mechanisms behind the modeled changes and in contextualizing the persistent model biases. The following specific comments aim to help the authors improve the manuscript.

We thank the reviewer for their constructive and insightful comments on our manuscript. We appreciate the recognition of the scientific relevance of our work, as well as the suggestions that help strengthen clarity, methodological transparency, and depth of interpretation. Below we respond to each point in detail and have revised the manuscript accordingly.

The description of the nudging technique is brief. Please specify the nudging strength (e.g., relaxation time scale) and the vertical extent above the PBL where it is applied. Furthermore, while you mention that nudging "can also dampen aspects of the fast response," a more detailed discussion is needed on how nudging might specifically affect the evaluation of cloud microphysical responses to your new parameterizations, which are process-based and potentially sensitive to synoptic variability.

We thank the reviewer for highlighting the need for a more detailed explanation of how nudging may affect the evaluation of cloud microphysical responses. In the revised manuscript, we now specify both the relaxation time scale of the nudging (3.33 hours) and the vertical extent where it is applied (from ~480 hPa, approximately 6450 m geopotential height, upward). We also clarify how nudging interacts with process-based parameterizations. In particular, nudging is applied only to wind divergence and vorticity, and not to temperature or specific humidity, so the model retains full control over thermodynamic fields that drive cloud microphysical processes. Constraining the large-scale flow can still damp fast dynamical fluctuations, which may modestly reduce short-term responses in cloud evolution. However, because nudging is partial and all simulations are nudged identically, internal dynamics, equilibrium, and relative differences between experiments remain interpretable. This clarification is now included in the revised methods section.

Original paragraph: "To facilitate direct comparison with observations and to isolate the short-term cloud responses under different parameterizations, we applied a nudging technique above the PBL. Specifically, wind divergence and vorticity fields were gently nudged towards ECMWF Reanalysis v5 (ERA5) data (Hersbach et al., 2017). While nudging helps reduce biases and drifts, improving the model's alignment with real-world conditions, it can also dampen aspects of the fast response. Nevertheless, comparisons between nudged and free-running simulations, included in the supplementary material (Figs. S1, S2 and S3), show minimal differences in zonal mean fields, supporting the robustness of our approach."

Revised paragraph: "To facilitate direct comparison with observations and to isolate short-term cloud responses under different parameterizations, we applied a nudging technique starting at ~480 hPa (approximately 6450 m geopotential height), which ensures that nudging is applied well above the planetary boundary layer (PBL). This prevents interference with PBL turbulence and surface fluxes while still constraining the large-scale flow. From this level upward, the wind divergence and vorticity fields were relaxed toward ECMWF Reanalysis v5 (ERA5) data (Hersbach et al., 2017) with a relaxation time scale of 3.33 hours. Nudging reduces dynamical drift and constrains the large-scale flow, but it does not apply to temperature or specific humidity, allowing the model to retain full control over the thermodynamic fields that drive cloud microphysical tendencies (e.g., growth rates of ice crystals and formation of mixed-phase clouds). Some fast dynamical fluctuations may be damped, potentially reducing the absolute amplitude of short-term cloud responses; however, because all configurations are nudged identically, relative differences between parameterizations remain interpretable. Comparisons between nudged and free-running simulations, included in the supplementary material (Figs. S1, S2, and S3), show minimal differences in zonal-mean fields, supporting the robustness of our approach."

The choice of a 12-year simulation period (2009-2020) is justified for satellite overlap. However, for INP evaluation which uses specific campaign data, is this period representative of the climatological INP state, or could interannual variability (e.g., in dust emission or marine productivity) influence the comparison scores (e.g., P_t1)? A brief comment on this would be helpful.

We thank the reviewer for this insightful comment. While the 12-year simulation period (2009–2020) was initially selected to ensure maximum overlap with satellite datasets, it also provides a reasonable basis for representing climatological INP conditions. Regarding the comparison with observational datasets, we distinguish between two categories:

For older measurements, such as those from Bigg (1973, 1990), we perform climatological comparisons using long-term monthly means. While these datasets cannot capture interannual variability, they provide broad spatial coverage—including regions where recent or updated INP measurements are still lacking. As such, they provide a valuable reference for assessing the spatial distribution of INP on a climatological scale.

For more recent field campaigns or continuous measurements, we use temporally and spatially co-located comparisons with the model output. This approach enables a more direct evaluation that inherently accounts for short-term variability in factors such as dust emissions, marine productivity, and other sources influencing INP concentrations. In

particular, we include continuous field measurements that capture interannual variability, such as the study by Wex et al. (2019), which reported INP concentrations from 2015–2017 at multiple sites across the Northern Hemisphere, including Ny-Ålesund, Svalbard. These datasets are well aligned with our simulation period and provide valuable constraints for evaluating model performance under real-world variability.

We acknowledge that interannual variability can affect pointwise metrics such as P_t1 (defined as the percentage of model–observation pairs within one order of magnitude). However, the focus of our analysis is on the structural performance of different INP parameterizations. The 12-year simulation period helps reduce the influence of single-year anomalies and supports a more robust assessment of model skill.

Accordingly, we have added the following paragraphs to the manuscript. In the "Methodology" section (line 286 of the original manuscript):
"In evaluating model performance, we distinguish between long-term climatological datasets (e.g., Bigg, 1973, 1990) and more recent field campaigns. Older datasets provide broad spatial coverage but lack interannual variability, while recent multi-year time series (e.g., Wex et al., 2019) allow for spatially and temporally co-located comparisons with the model results. Given the 12-year simulation period (2009–2020), short-term variability in INPs is partly smoothed, helping to reduce sensitivity to single-year anomalies in pointwise metrics such as P_t1".

And in the "Results and discussion" section ( line 336 of the original manuscript), we have added:
"These evaluation scores reflect a combination of climatological comparisons (for older datasets) and co-located comparisons for recent field campaigns (Sect. 2.3), providing a consistent basis for assessing the spatial and temperature-dependent behavior of the INP schemes."

Bibliography added to the manuscript:
Bigg, E. K.: Ice Nucleus Concentrations in Remote Areas, J. Atmos. Sci., 30, 1153–1157, https://doi.org/10.1175/1520- 0469(1973)030<1153:INCIRA>2.0.CO;2, 1973.

Bigg, E. K.: Long-term trends in ice nucleus concentrations, Atmos. Res., 25, 409–415, https://doi.org/10.1016/0169-8095(90)90025-8, 1990.

Wex, H., Huang, L., Sheesley, R., Bossi, R., and Traversi, R.: Annual concentrations of ice nucleating particles at different Arctic stations, 925
PANGAEA [data set], https://doi.org/10.1594/PANGAEA.899701, 2019.

You state that riming tendencies are not diagnosed and are internally estimated by RaFSIP. This is a crucial detail. The reader needs a clearer understanding of how this internal estimation works, as riming is a key driver for several SIP mechanisms. The reference to Frostenberg et al. (2025) is given, but since this is a "to be submitted" reference, the manuscript should include a more self-contained, concise explanation of the method or its core assumptions in the main text or a dedicated appendix.

We thank the reviewer for highlighting the importance of clarifying the internal estimation of riming tendencies within RaFSIP. We have added a concise description in the manuscript and provided a preprint reference for the detailed methodology (Frostenberg et al., 2025, https://www.researchsquare.com/article/rs-7785490/v1).

Within EC-Earth3-AerChem, riming rates are not explicitly parameterized. To address this, two Random Forest regressors (RFRs), *forestrimall* and *forestrimc*, are trained on a two-year dataset to diagnose riming rates RIMC (for cloud droplets) and RIMR (for rain). The RFRs use temperature and mass mixing ratios of relevant liquid and ice species as inputs. *Forestrimall* predicts RIMC and RIMR when all mass mixing ratios are nonnegative, while *forestrimc* is used when specific rain content is zero, predicting RIMC from the remaining fields. Predictions are constrained within observed physical bounds. These riming rates serve solely as diagnostic inputs to RaFSIP and do not participate directly in the model's conservation equations.

This description provides a self-contained explanation of how RaFSIP internally estimates riming tendencies to drive secondary ice production in EC-Earth3-AerChem.

Original paragraph: "[...] Since riming tendencies are not diagnosed in EC-Earth3-AerChem, RaFSIP internally estimates them based on the predicted cloud and rain mass mixing ratios at each time step (see Frostenberg et al. (2025) for details on the implementation in EC-Earth, in particular Section A.2, which describes the random forest used to predict the riming tendencies). To prevent excessive SIP activity that could destabilize the model, we imposed upper bounds on the SIP-induced ICNC formation rates, with RaFSIPv1 capped at $100 \, kg^{-1} \, s^{-1}$, and RaFSIPv2 capped at $10 \, kg^{-1} \, s^{-1}$. These thresholds were established after extensive testing to ensure numerical stability without afecting physical realism."

Revised paragraph: "[...] Since riming tendencies are not diagnosed in EC-Earth3-AerChem, RaFSIP internally estimates them based on the predicted cloud and rain mass mixing ratios at each time step (see Frostenberg et al. (2025) for details on the implementation in EC-Earth, in particular Section A.2, which describes the random forest used to predict the riming tendencies). Specifically, two Random Forest regressors (RFRs), *forestrimall* and *forestrimc*, are trained on a two-year dataset to diagnose riming rates RIMC (for cloud droplets) and RIMR (for rain). The RFRs use temperature and mass mixing ratios of relevant liquid and ice species as inputs. *Forestrimall* predicts RIMC and RIMR when all mass mixing ratios are nonnegative, while *forestrimc* is used when specific rain content is zero, predicting RIMC from the remaining fields. Predictions are constrained within observed physical bounds and serve solely as diagnostic inputs to RaFSIP; they do not participate directly in the model's conservation equations. Finally, to prevent excessive SIP activity that could destabilize the model, we imposed upper bounds on the SIP-induced ICNC formation rates, with RaFSIPv1 capped at $100 \, kg^{-1} \, s^{-1}$, and RaFSIPv2 capped at $10 \, kg^{-1} \, s^{-1}$. These thresholds were established after extensive testing to ensure numerical stability without affecting physical realism."

Lines 340-345: The caveat regarding surface vs. aloft INP dominance is critical. This point should be elevated and expanded upon in the discussion (Section 4). What are the implications for your model's cloud impacts if mineral dust dominates ice nucleation aloft in

many regions, while your combined scheme's superior performance is primarily driven by better surface INP representation from marine organics? Does this create a potential disconnect between INP evaluation skill and actual cloud impact skill?

We thank the reviewer for highlighting this important point regarding the potential disconnect between surface INP evaluation and cloud impacts aloft. As noted in the manuscript, indeed most INP observations are collected near the surface, so good agreement there does not automatically guarantee accurate representation of cloud microphysics at higher altitudes. This disconnect is a well-known and pervasive limitation in current INP evaluations, not only in our study but across the field. At higher altitudes, mineral dust likely remains the dominant INP species in many regions, influencing ICNC and cloud-phase partitioning in mid- and upper-tropospheric mixed-phase clouds. To address this, we explicitly evaluate cloud microphysical variables (e.g., cloud cover, LWP, IWP) across the different parameterizations, ensuring that the configuration with the highest surface INP skill also produces realistic cloud responses aloft. We have expanded the discussion in Section 4 to highlight this caveat and its implications, which are also important for guiding future observational campaigns targeting upper-tropospheric INPs.

Correspondingly, in the conclusions section, we have added the following clarifications:

[...] The aerosol-sensitive scheme leads to improved agreement with global in-situ observations of ice nucleating particles (INPs), particularly when K-feldspar, quartz and marine organics are included. The resulting spatial distribution of INPs is more realistic than that produced by the traditional temperature-only scheme, especially over remote oceans and regions with minimal dust input. However, since most INP observations are near the surface, good agreement there does not guarantee accurate cloud microphysics aloft; this disconnect between surface INP evaluation skill and cloud impact skill is a general limitation affecting current modelling studies. Additional measurements in the upper troposphere would be valuable for further constraining model performance and guiding future observational campaigns targeting upper-tropospheric INPs. To address this, we also evaluate cloud variables (e.g., cloud cover, LWP, IWP), ensuring that the configuration with the highest surface INP skill also produces realistic cloud responses at higher altitudes. Adding the SIP scheme—especially the more advanced RaFSIPv2—amplies ICNC in regions with low primary INPs, such as the Southern Ocean, helping to rebalance the cloud phase.
In terms of cloud macrophysical properties, the new parameterizations increase cloud cover and LWP in mid- and high latitude regions, reflecting greater persistence of supercooled liquid water. [...]

Lines 360-370: The explanation for the limited impact of RaFSIPv1 is clear. However, the description of the "forestBRwarm" mechanism in RaFSIPv2 and other "additional pathways" remains vague. Please provide a more concrete, physical description of the primary mechanisms that make RaFSIPv2 effective independent of concurrent PIN. For example, how does the scheme handle seeder-feeder processes conceptually?

We thank the reviewer for requesting a clearer explanation of the additional SIP pathways represented in RaFSIPv2, and of how the scheme handles processes such as seeder–feeder interactions. In the revised manuscript, we have expanded the description of

the Random Forest regressors (RFRs) and the physical mechanisms they encode in section 2.2.3.

RaFSIPv2 differs fundamentally from RaFSIPv1 in that SIP is no longer directly tied to the local primary ice production (PIN) rate. RaFSIPv2 uses multiple RFRs that activate under distinct microphysical and thermodynamic conditions (see section "2.2.3 Secondary ice production schemes" and Table 3 in Frostenberg et al., 2025). This structure allows the scheme to respond to environmental drivers of SIP even when primary INPs are scarce.

A key addition in RaFSIPv2 is the *forestBRwarm* RFR, which predicts ice-ice collisional break-up (BR; Vardiman, 1978; Takahashi et al., 1995) at warm subzero temperatures (−3 to 0 °C). As noted by Georgakaki and Nenes (2025), droplet shattering is unlikely to operate efficiently in this temperature range outside of deep convective or frontal clouds; therefore, BR initiated by rimed seeding particles is expected to be the dominant SIP pathway. Importantly, *forestBRwarm* is sensitive to increases in ice water content (IWC) caused by ice sedimenting from higher-level clouds, a process repeatedly identified as a key driver of BR (Ramelli et al., 2021; Georgakaki et al., 2022; Järvinen et al., 2022; Pasquier et al., 2022; Sotiropoulou et al., 2021). This allows RaFSIPv2 to represent a seeder-feeder-type mechanism, in which falling ice enhances SIP when interacting with supercooled droplets in lower-level mixed-phase clouds.

Takahashi, T., Nagao, Y., & Kushiyama, Y. (1995). Possible high ice particle production during Graupel–Graupel collisions. Journal of the Atmospheric Sciences, 52(24), 4523–4527. https://doi.org/10.1175/1520-0469(1995)052<4523:phippd>2.0.co;2

Added paragraph in section 2.2.3:

"In RaFSIPv2, the extension of collisional breakup up to 0 °C is achieved through a key addition, the *forestBRwarm* random forest regressor, which predicts ice-ice collisional break-up (BR) at warm subzero temperatures (−3 to 0 °C). In this regime, droplet shattering is generally inefficient outside deep convective or frontal systems (Georgakaki and Nenes, 2024), making BR initiated by rimed ice particles the dominant SIP mechanism. forestBRwarm can also respond to increases in ice water content (IWC) associated with ice sedimenting from higher-level clouds, a process frequently identified as a driver of BR (Ramelli et al., 2021; Sotiropoulou et al., 2021; Georgakaki et al., 2022; Järvinen et al., 2022; Pasquier et al., 2022), thus allowing RaFSIPv2 to conceptually mimic a seeder-feeder-type enhancement in mixed-phase cloud regimes."

New references added to the manuscript:

Pasquier, J. T., Henneberger, J., Ramelli, F., Lauber, A., David, R. O., Wieder, J., et al. (2022). Conditions favorable for secondary ice production in Arctic mixed-phase clouds. Atmospheric Chemistry and Physics, 22(23), 15579–15601. https://doi.org/10.5194/acp-22-15579-2022

Ramelli, F., Henneberger, J., David, R., Bühl, J., Radenz, M., Seifert, P., et al. (2021). Microphysical investigation of the seeder and feeder region of an Alpine mixed-phase cloud. Atmospheric Chemistry and Physics, 21(9), 6681–6706. https://doi.org/10.5194/acp-21-6681-2021

Sotiropoulou, G., Ickes, L., Nenes, A., and Ekman, A. M. L. (2021): Ice multiplication from ice–ice collisions in the high Arctic: sensitivity to ice habit, rimed fraction, ice type and uncertainties in the numerical description of the process, Atmos. Chem. Phys., 20, 9741–9760, https://doi.org/10.5194/acp-21-9741-202

Lines 420-450: The discussion of persistent LWP/IWP biases and the role of COSP simulator artifacts is excellent and honest. However, the conclusion that "the main structural biases are not removed by representing PIN and SIP" is somewhat buried. This should be a central, clearly stated takeaway in the abstract and conclusions.

We thank the reviewer for highlighting the need to clearly emphasize that the main structural cloud biases are not removed by representing PIN and SIP. However, this point is already addressed in the conclusions (Lines 565-569):

> "While the aerosol-sensitive scheme introduces physically grounded improvements and allows for more nuanced diagnostics of aerosol–cloud interactions, it does not resolve all known cloud and radiation biases in EC-Earth3. Many of these shortcomings appear to originate from the structural formulation of cloud and convection parameterizations in IFS Cy36R4, for example, the hard-coded thresholds for phase transitions and simplified detrainment assumptions–rather than from the treatment of ice nucleation itself."

This statement makes explicit that, despite improvements in cloud phase and ICNC from aerosol-sensitive PIN and SIP, the primary structural LWP/IWP biases persist, which was not expected to be solved by these parameterizations.

We have revised the abstract to make this point clearly highlighted:

[...] The new configuration improves agreement with global in situ ice nucleating particle (INP) observations and reveals realistic spatial patterns of ice crystal number concentrations (ICNC) across diverse environments. While these improvements do not eliminate the persistent structural cloud biases in EC-Earth3-AerChem, the aerosol-sensitive primary ice production scheme increases supercooled liquid water and cloud cover, particularly in the extratropics. Critically, the addition of SIP rebalances the cloud phase by enhancing ICNC in regions with low primary ice formation.[...]

Lines 490-510: The analysis of radiative effects is comprehensive. However, the interpretation of the "net warming" introduced by RaFSIPv2 (Fig. 5i) needs strengthening. Is this warming primarily due to LWP decrease (as stated) leading to reduced cloud albedo? Link this mechanism more explicitly to the physical processes in SIP (e.g., conversion of liquid to ice via rime splintering, potentially leading to faster precipitation and reduced cloud lifetime/albedo). Also, discuss the climatic significance of a +10 W/m² TOA CRE anomaly at high southern latitudes.:

We have revised the text to provide a clearer mechanistic explanation linking RaFSIPv2's net warming to enhanced SIP-mediated liquid–ice conversion, which shifts the cloud phase and reduces LWP. We now explicitly connect these microphysical processes to the observed LWP reduction and discuss the climatic relevance of a regional +10 W/m² TOA CRE anomaly, particularly in the Southern Ocean.

Original paragraph:

"The isolated impact of SIP (both with RaFSIPv1 and RaFSIPv2) is shown in the third column of Fig. 5. Including RaFSIPv1 has an insignificant effect on the cloud properties and therefore has little impact on the CRE. In contrast, including RaFSIPv2 results in a net warming of the system mainly due to a decrease in LWP, as indicated by a positive net CRE at the TOA (Fig. 5i). This net warming is particularly pronounced at high southern latitudes (~60º S). At these latitudes, the shortwave TOA CRE increases by approximately 13 W/m², while the longwave TOA CRE decreases by about 3 W/m², yielding a net TOA CRE anomaly up to +10 W/m². At the equator, the net effect is about +5 W/m², and across the mid-latitudes in the NH (30–60°N), the net TOA CRE increase is approximately +6 W/m²."

Revised paragraph:

"The isolated impact of SIP (both with RaFSIPv1 and RaFSIPv2) is shown in the third column of Fig. 5. Including RaFSIPv1 has an insignificant effect on the cloud properties and therefore has little impact on the CRE. In contrast, including RaFSIPv2 results in a net warming of the system mainly due to a decrease in LWP, as indicated by a positive net CRE at the TOA (Fig. 5i). This LWP reduction is consistent with SIP-induced ice multiplication (e.g., rime-splintering), which converts supercooled liquid water into ice, thereby shifting the cloud phase from liquid to ice. This net warming is particularly pronounced at high southern latitudes (~60° S), where the shortwave TOA CRE increases by approximately 13 W/m², while the longwave TOA CRE decreases by about 3 W/m², yielding a net TOA CRE anomaly up to +10 W/m². At the equator, the net effect is about +5 W/m², and across the mid-latitudes in the NH (30–60°N), the net TOA CRE increase is approximately +6 W/m². This +10 W/m² TOA CRE anomaly at high southern latitudes indicates a substantial regional radiative impact, highlighting the importance of SIP in modulating cloud albedo and the regional energy balance, particularly in regions with low primary ice formation such as the Southern Ocean."

We thank the reviewer again for the detailed and thoughtful comments. Their suggestions have substantially improved the clarity, physical interpretation, and overall scientific rigour of the manuscript.